# Mini-bacterioferritins: structural insight into a ferritin-like protein from the anaerobic methane-oxidising archaeon *Candidatus* Methanoperedens carboxydivorans

Martijn Wissink [1], Sylvain Engilberge [2], Pedro Leão [1], Robert S. Jansen [1], Mike S. M. Jetten[1], Mélissa Belhamri[2], Olivier N. Lemaire[2], Antoine Royant [2,3], Cornelia U. Welte [1,5] ✉ & Tristan Wagner [2,4,5] ✉

Ferritins are ubiquitous among life forms, as they are essential for iron homeostasis. Here, we unveiled a novel member of the ferritin family, baptised mini-bacterioferritin. The characterised mini-bacterioferritin was isolated from a microbial enrichment dominated by the methanotrophic archaeon '*Candidatus* Methanoperedens carboxydivorans'. Its atomic resolution crystal structure reveals a 12-mer assembly with a diiron ferroxidase centre located within a four-helix bundle. Redox-cycling experiments on protein crystals reveal a shift in iron position at the active site, which follows the established ferritin catalytic cycle. The 12-mer sphere-like structure harboured six Fe-coproporphyrin III ligands, positioned at the interdimeric interface, a characteristic previously only found in 24-mer bacterioferritins. Phylogenetics, together with structure predictions of closely related proteins, revealed that mini-bacterioferritins form a distinct clade within the ferritin family that might conserve ancestral traits. Future research will need to investigate the physiological roles of these enzymes, which were unsuspectingly widely distributed among prokaryotes.

Iron is essential for Life, yet iron homeostasis is challenging due to the low solubility of $Fe^{3+}$ and the toxicity of $Fe^{2+}$[1]. To address these challenges, all domains of Life rely on proteins belonging to the ferritin family that store iron in a non-harmful way inside the cell[2]. The ferritin family consists of four groups: ferritins, bacterioferritins, 'DNA-binding proteins from starved cells' (Dps) and Dps-like proteins (DpsL)[3,4]. The defining feature of ferritin-like proteins is the conserved four-helix bundle domain, with helices A and B antiparallel to C and D, connected by a long loop between helices B and C[5]. Despite high structural similarities within ferritin-like proteins, amino acid sequences vary, with similarities as low as 20%[6,7].

Ferritins oligomerise as 24-mer sphere-like structures[2,3,7–9]. While ferritins are composed of heavy and light chains in eukaryotes, they generally are homomultimers in prokaryotes, with each subunit containing a nuclear diiron ferroxidase centre that oxidises $Fe^{2+}$ to $Fe^{3+}$ using either oxygen or hydrogen peroxide[9,10]. The iron is then mineralised and stored inside the hollow spherical cavity of the protein, which can hold up to 4500 iron atoms[11]. When iron is scarce, the stored iron is released from the ferritin to the cytosol for incorporation into iron-containing proteins.

24-mer bacterioferritins, found exclusively in prokaryotes, are structurally and functionally similar to ferritins[5,6]. The primary distinction is that bacterioferritins bind 12 haems, which are believed to act as electron conduits, facilitating the reduction of $Fe^{3+}$ within the protein via bacterioferritin-associated ferredoxin (Bfd)[12]. Typically, bacterioferritins contain haem *b*, although there is one example of a bacterioferritin from *Desulfovibrio desulfuricans* (*Dd*-Bfr) that contains a Fe-coproporphyrin-III (coproheme) instead[13,14].

[1]Department of Microbiology, Radboud Institute for Biological and Environmental Sciences, Radboud University, Nijmegen, the Netherlands. [2]Institut de Biologie Structurale, CEA, CNRS, Université Grenoble Alpes, Grenoble, France. [3]European Synchrotron Radiation Facility, Grenoble, France. [4]Microbial Metabolism Research Group, Max Planck Institute for Marine Microbiology, Bremen, Germany. [5]These authors jointly supervised this work. ✉e-mail: c.welte@science.ru.nl; tristan.wagner@ibs.fr

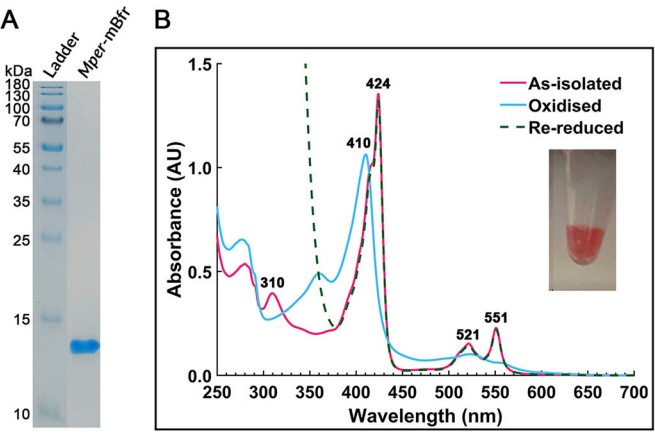

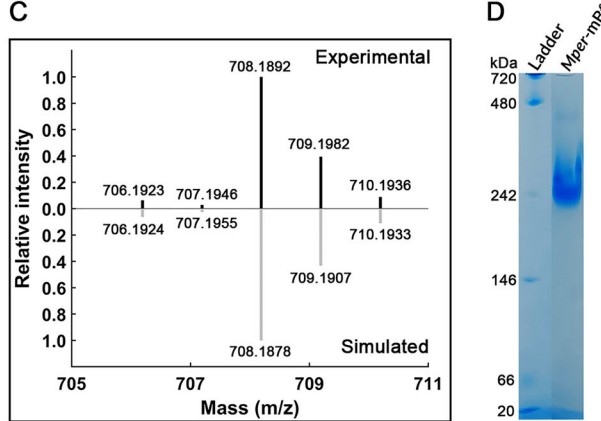

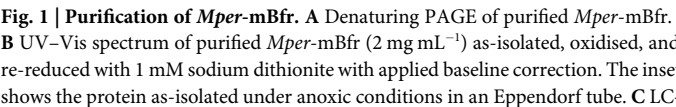

**Fig. 1 | Purification of *Mper*-mBfr. A** Denaturing PAGE of purified *Mper*-mBfr. **B** UV–Vis spectrum of purified *Mper*-mBfr (2 mg mL$^{-1}$) as-isolated, oxidised, and re-reduced with 1 mM sodium dithionite with applied baseline correction. The inset shows the protein as-isolated under anoxic conditions in an Eppendorf tube. **C** LC-MS spectrum of *Mper*-mBfr extracted haems versus the simulated MS1 spectrum of $C_{36}H_{36}FeN_4O_8$. **D** hrCN-PAGE of purified *Mper*-mBfr. **A, D** The ladder and the loaded sample were originally not adjacent. See Supplementary Fig. 2 for original images.

'DNA-binding proteins from starved cells' (Dps) differ from ferritins and bacterioferritins in several ways, but most notably by forming 12-mer complexes instead of 24-mer complexes[15]. As in the case of (bacterio)ferritins, Dps can store iron, and their primary role is to defend cells against a wide range of stresses, including oxidative stress, metal toxicity, thermal stress, and starvation[16]. Apart from ferroxidase activity, Dps exhibit additional properties such as DNA binding, endonuclease activity, and protease activity[17,18]. Unlike ferritins and bacterioferritins, Dps have their active site at the interface between two subunits, where two iron atoms are coordinated by ligands from both subunits[19].

The most recently characterised group within the ferritin family is the Dps-like (DpsL) group. Structurally similar to Dps, DpsL also form 12-mer assemblies, but its ferroxidase active site more closely resembles that of ferritins and bacterioferritins[20–22]. Phylogenetically, DpsL are more closely related to bacterioferritins than to Dps[3,4]. Like Dps, transcription of the gene encoding for DpsL is upregulated upon oxidative stress, and DpsL is often associated with additional functions, such as endonuclease activity[4,21].

To hunt for unexplored ferritin systems used in microbial life, we looked for ferritin-like proteins with sequence identity divergent from known homologues and focused on 'Candidatus Methanoperedens carboxydivorans'[23] (formerly 'Candidatus Methanoperedens' BLZ2[24]). The genome of this archaeon harbours five ferritin-like proteins (Supplementary data and Supplementary Fig. 1), and two of them present naturally high expression as shown by a complexome profiling study[25]. 'Ca. M. carboxydivorans' is an anaerobic methane-oxidising archaeon that thrives in anoxic environments, in particular freshwater sediments, where it contributes to the biogeochemical methane cycle by coupling the oxidation of methane to the reduction of nitrate and extracellular electron acceptors such as iron and manganese[26–28]. To avoid any biases from recombinant expression (e.g. determining if the protein contains a particular intrinsic cofactor), we sought to natively purify the most abundant ferritin-like protein from this organism.

With this successful approach, we solved the structure of the first representative of a previously unrecognised group within the ferritin family. The atomic resolution structure supported by biochemical studies reveals a 12-mer sphere-like structure harbouring two redox-active catalytic iron per monomer and a total of six coprohemes. Placing the 'Ca. M. carboxydivorans' ferritin-like protein in a phylogenetic tree showed the existence of similar proteins that form a distinct clade within the ferritin family.

## Results
### Isolation of mini-bacterioferritin from 'Ca. M. carboxydivorans'
Anaerobic methane-oxidising archaea (ANME) present significant research challenges due to their lack of axenic cultures, slow growth rates (doubling time ~30 days in bioreactors), and sensitivity to oxygen. Nevertheless, enrichment cultures of ANME species have enabled the study of their physiology and biochemistry. Here, the enrichment culture of 'Ca. M. carboxydivorans' was maintained for over a decade in a bioreactor fed with methane as the sole electron donor and nitrate as the electron acceptor. This setup produced a stable community with 20–40% enrichment of 'Ca. M. carboxydivorans' that could be exploited for native purification.

Following a strict anaerobic purification under yellow light, we enriched a ferritin-like protein with an intense pink colour after three chromatography steps. Starting from 17.5 g wet weight biomass, the process yielded approximately 1 mg purified protein. The final purified fraction (with >95% purity confirmed by denaturing polyacrylamide gel electrophoresis, PAGE, Fig. 1A and Supplementary Fig. 2) was identified by mass spectrometry. This protein of 139 amino acids (15.9 kDa) corresponds to a naturally abundant putative bacterioferritin found in meta-proteomics data[25] (Fig. 1). The UV–Visible spectrum of the protein confirmed a haem presence, with a distinct Soret peak at 424 nm with a shoulder at 418 nm, and Q bands at 521 and 551 nm resembling the haem spectrum of *Dd*-Bfr[29] (Fig. 1B). A small peak at 310 nm is detected in the as-isolated form (Fig. 1B), however, it is highly improbable that this peak originates from Fe(III)-O- moieties in the interior cavity since the presence of Fe(III)-O- is expected to give rise to a broad intense band near 300 nm[29–32]. Furthermore, this peak disappears in the presence of oxygen, providing further support that it does not originate from Fe(III)-O-, and would rather be a feature of the Fe-porphyrin system[29–32]. Mass spectrometry (MS) of the extracted haem yielded a compound with a $m/z$ of 708.189 [M + H]$^+$, which was predicted to have an elemental composition of $C_{36}H_{36}FeN_4O_8$ based on MS and MS/MS spectra (Fig. 1C). In silico structure prediction using CANOPUS software classified the compound within the porphyrin subclass. Previous work showed that coproheme was detected at $m/z$ 708 using LC-MS[33]. Together, these data strongly suggest that the haem extracted from *Mper*-mBfr is a coproheme. Spectral shifts upon air oxidation and dithionite reduction confirmed reversible redox activity, consistent with a coordinated redox-active haem (Fig. 1B).

The oligomerisation state of the protein was investigated by high-resolution clear native PAGE (hrCN-PAGE) under anoxic conditions. It revealed a protein assembly of approximately 242 kDa (Fig. 1D). While ferritin-family proteins are known to assemble into larger quaternary structures, their size was smaller than expected for a canonical 24-subunit bacterioferritin (i.e. expected to be 381.6 kDa). Based on its origin and characteristics, the protein was provisionally named *Mper*-mBfr (*Methanoperedens* mini-bacterioferritin). Its unexpected deviation in size and sequence from typical bacterioferritins prompted structural investigation, and anaerobic protein crystallisation was undertaken.

https://doi.org/10.1038/s42003-026-09796-4     **Article**

**Table 1 | X-ray analysis statistics for *Mper*-mBfr**

| | *Mper*-mBfr as-isolated | *Mper*-mBfr without treatment | *Mper*-mBfr O₂ exposed | *Mper*-mBfr O₂ exposed, then DT reduced |
|---|---|---|---|---|
| **Data collection** | | | | |
| Synchrotron and beamline | SLS, PXI-X06SA | ESRF, BM07-FIP2 | ESRF, BM07-FIP2 | ESRF, BM07-FIP2 |
| Wavelength (Å) | 1.00000 | 0.97930 | 0.97930 | 0.97930 |
| Space group | $P6_3$ | $P2_13$ | $P2_13$ | $P2_13$ |
| Resolution (Å) | 75.87–1.07 (1.13–1.07) | 86.73–1.68 (1.70–1.68) | 86.80–1.54 (1.56–1.54) | 86.90–2.08 (2.11–2.08) |
| **Cell dimensions** | | | | |
| *a, b, c* (Å) | 87.61, 87.61, 131.37 | 122.66, 122.66, 122.66 | 122.75, 122.75, 122.75 | 122.90, 122.90, 122.90 |
| *α, β, γ* (°) | 90, 90, 120 | 90, 90, 90 | 90, 90, 90 | 90, 90, 90 |
| $R_{merge}$ (%)[a] | 8.2 (105.2) | 20.6 (371.9) | 15.8 (185.1) | 17.4 (129.5) |
| $R_{pim}$ (%)[a] | 2.0 (46.6) | 5.8 (104.8) | 5.1 (83.0) | 6.4 (92.5) |
| $CC_{1/2}$[a] | 0.999 (0.654) | 0.996 (0.323) | 0.997 (0.371) | 0.994 (0.374) |
| $I/\sigma_I$[a] | 18.5 (1.7) | 8.8 (0.8) | 7.5 (0.7) | 7.1 (0.7) |
| Spherical completeness[a] | 87.9 (28.4) | 98.6 (100.0) | 100.0 (100.0) | 99.7 (98.1) |
| Ellipsoidal completeness[a] | 95.8 (58.4) | / | / | / |
| Redundancy[a] | 15.7 (5.9) | 13.4 (13.5) | 10.2 (5.8) | 7.7 (2.7) |
| Nr. unique reflections[a] | 221,484 (11,074) | 69,454 (3470) | 91,535 (4547) | 37,379 (1806) |
| **Refinement** | | | | |
| Resolution (Å) | 75.87–1.07 | 38.79–1.68 | 38.82–1.54 | 86.90–2.08 |
| Number of reflections | 221,476 | 69,258 | 91,188 | 36,283 |
| $R_{work}/R_{free}$[b] (%) | 10.78/12.54 | 16.22/18.11 | 16.55/18.16 | 19.55/22.95 |
| **Number of atoms** | | | | |
| Protein | 9583[d] | 4499 | 4574 | 4526 |
| Ligands/ions | 450[d] | 212 | 216 | 205 |
| Solvent | 924 | 867 | 838 | 540 |
| Mean *B*-value (Å²) | 13.31 | 28.19 | 24.00 | 30.37 |
| Molprobity clash score | 0.10 | 0.11 | 0.32 | 0.33 |
| **Ramachandran plot** | | | | |
| Favoured regions (%) | 99.45 | 98.89 | 98.89 | 99.08 |
| Outlier regions (%) | 0 | 0 | 0 | 0 |
| rmsd[c] bond lengths (Å) | 0.010 | 0.007 | 0.007 | 0.006 |
| rmsd[c] bond angles (°) | 1.267 | 0.869 | 0.890 | 0.822 |
| PDB code | 9QQ4 | 9QQI | 9QQ5 | 9QQH |

[a]Values relative to the highest resolution shell are within parentheses.
[b]$R_{free}$ was calculated as the $R_{work}$ for 5% of the reflections that were not included in the refinement.
[c]rmsd, root mean square deviation.
[d]Model contains hydrogens.

## *Mper*-mBfr forms a 12-mer sphere

*Mper*-mBfr crystallised into bright pink bipyramidal crystals within two weeks. X-ray diffraction data were collected, and the structure was refined to an atomic resolution of 1.07 Å (Table 1). Structural analysis revealed a 12-mer assembly with a 23 tetrahedral symmetry (Fig. 2A), resembling that of Dps[34].

In this assembly, the N-terminal ends of three four-helix bundles are facing each other and form the N-terminal pore. Conversely, in the other conformation, the C-terminal ends are facing each other and form the C-terminal pore (Fig. 2A). Since the 432 octahedral symmetry of (bacterio) ferritins only allows for N-terminal-type pores, these pores are also referred to as ferritin-type pores, while C-terminal-type pores are referred to as Dps-type pores[34]. Each monomer contains a diiron active site, and one copro-heme is bound at the interdimeric interface (Fig. 2).

*Mper*-mBfr forms a hollow spherical structure with an outer diameter of ~90 Å and an inner diameter of ~46 Å (Fig. 2B), with dimensions comparable to *Escherichia coli* Dps (*Ec*-Dps) (~90 Å and ~45 Å, respectively) but smaller than *Dd*-Bfr (~130 Å and ~85 Å, respectively). The inner volume of *Mper*-mBfr was estimated to be 58 nm³ (see 'Materials and Methods' section for volume calculations), comparable to that of Dps and DpsL, such as *Halobacterium salinarum* Dps (*Hs*-Dps) (56 nm³)[35] and *Pseudomonas aeruginosa* DpsL (*Pa*-DpsL) (75 nm³)[21]. In contrast, *Dd*-Bfr has a much larger inner volume of 251 nm³[13].

## Electrostatic surface potential of *Mper*-mBfr

The outer electrostatic surface potential of ferritin-like proteins plays a crucial role in metal ion transport through pores, and in the case of Dps, in their ability to bind DNA. While the outer charge profile of *Mper*-mBfr reveals interspersed regions of localised positive and negative charges (Fig. 3), the inner surface presents defined positive patches for haem binding, with the rest being negatively charged, creating a suitable environment for storing positively charged iron cations as observed in *Ec*-Dps and *Dd*-Bfr[13,34].

In ferritin family proteins, three-fold pores have been associated with metal ion transport, typically facilitated by a pathway of negatively charged amino acids[13,20,34,36]. In *Ec*-Dps and three types of *Nostoc punctiforme* Dps, the N-terminal pore exhibits a significantly negative potential, while the C-terminal pore remains electrostatically neutral. In contrast, *Mper*-mBfr displays the opposite pattern, with more negative electrostatic potential around the C-terminal pore and a relatively neutral N-terminal pore. This electrostatic distribution resembles what has been observed in *Saccharolobus solfataricus* DpsL and *Dd*-Bfr, suggesting that the C-terminal pore is the more likely route for iron transport into the protein cavity.

## Dual conformation of coprohemes at the dimeric interface

The coproheme is axially coordinated by Met50 located at the dimeric interface along the 2-fold symmetry axis. The atomic resolution electron density map revealed a dual conformation of the coproheme ligand, with each orientation modelled at 50% occupancy, fitting at best the individual atomic B-factor profile (Fig. 4). The same dual coproheme-binding alternative conformation has been reported in the coproheme binding *Dd*-Bfr, but also for *Rhodobacter capsulatus* bacterioferritin that binds haem *b*[13,37].

The coproheme is positioned within a hydrophobic pocket, where its two internally facing propionate arms are stabilised by hydrogen bonds with Arg13 and Tyr28 from both amino acid chains in the dimer (Fig. 4B), an interaction similar to that observed in *Dd*-Bfr. In contrast, the two solvent-exposed propionate arms extend toward the interior of the spherical assembly, but their lack of clear electron density reveals high flexibility, supported by the B-factor profile (Fig. 4C). The coproheme is accessible only from the interior and remains shielded from the external solvent (Fig. 3), a feature consistent with other bacterioferritins. Interestingly, in *Dd*-Bfr, the outward-facing propionate arms are not solvent-exposed but are further stabilised by interactions with C-terminal residues[13].

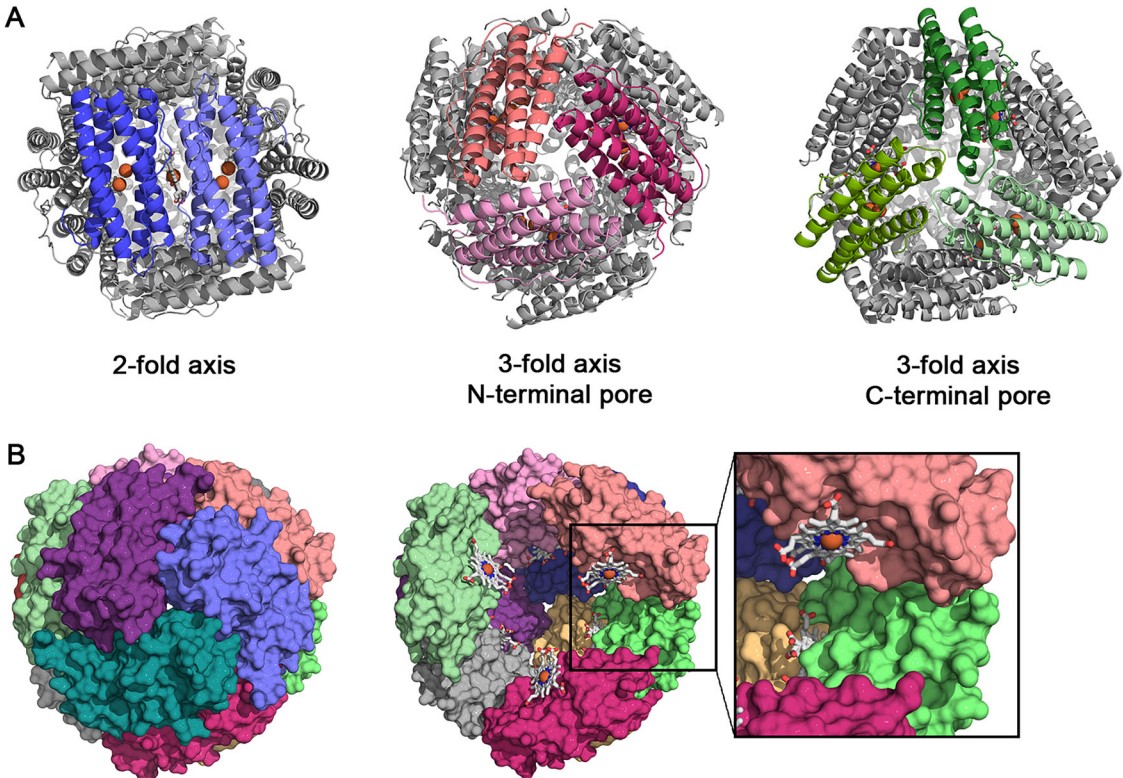

**Fig. 2 | 12-mer structure of *Mper*-mBfr. A** Symmetry interfaces of *Mper*-mBfr are shown as cartoons and viewed along the 2-fold symmetry axis, 3-fold symmetry axis viewing the N-terminal pore, and the 3-fold symmetry axis viewing the C-terminal pore. **B** Surface representation of *Mper*-mBfr (left) and surface view without the front chains (coloured purple, slate blue and deep teal), showing the coordination of haem ligands (right). Haem ligands are represented as balls and sticks and iron atoms as spheres, with carbons, oxygen, nitrogen and iron coloured as white, red, dark blue and orange, respectively.

## The diiron ferroxidase centre is redox active *in crystallo*

The active site of *Mper*-mBfr conserved the canonical features of a bacterioferritin: two iron atoms coordinated by four glutamate residues (Glu16, Glu49, Glu91 and Glu124) and two histidine residues (His52 and His127) within the characteristic four-helix bundle (Fig. 5). Additionally, the Glu16 and Glu91 are stabilised by hydrogen bonding with Tyr23 and Tyr98. Each iron atom is hexacoordinated by the mentioned residues and a water molecule (Fig. 5), and separated by a distance of 3.9 Å, typical for a reduced ferrous state[13].

To investigate whether the ferroxidase centre of *Mper*-mBfr is redox active, protein crystals were subjected to an oxidation-reduction cycle. *In crystallo* UV–Vis absorption spectroscopy was performed to corroborate structural alterations induced by the oxidation from $O_2$ exposure and back-reduction provoked by the dithionite agent. A control crystal representing the as-isolated state exhibited a spectrum with sharp peaks at 521 and 551 nm under cryogenic conditions, which was comparable to the solution spectrum at room temperature, confirming that the protein was in its reduced state, as previously observed (Figs. 5 and 6A). Upon oxidation by ambient air, a decrease in absorbance was observed. The decrease was more pronounced at 551 nm than at 521 nm. The latter is also getting wider. A second $O_2$-exposed crystal that underwent the same colour change could be reduced back by soaking in dithionite, which restored the spectrum to its as-isolated state (Fig. 6A and Supplementary Fig. 3).

The three redox captured states did not show any major shift in the protein backbone with a root mean square deviation below 0.1 Å for around 132 Cα (Fig. 6B). A comparison of the ferroxidase centres (Fig. 6C–E and Supplementary Fig. 4) reveals a partial shift of Fe2 moving away from His127 and a slight twist of Glu49 in the oxidised state. The re-reduced state presents an identical coordination to the as-isolated state, with the Fe2 back into coordination with His127. A previous study on *Dd*-Bfr found Fe1 instead of Fe2 to be mobile upon redox cycling of protein crystals[13]. This study also showed that redox cycling of *Dd*-Bfr in solution prior to crystallisation resulted in complete loss of Fe1 and conformational changes in the bridging glutamates. Such a dramatic effect might have been prevented by the crystal packing or might have occurred if the crystals reached the fully oxidised state. A state that might have led to a loss of diffraction upon protein reorganisation.

The observation of the displacement of Fe2 upon redox cycling prompted us to investigate if *Mper*-mBfr harbours Fe-storing activity. For this, we performed Ferene S and Prussian Blue staining on hrCN-PAGE to detect Fe- and Fe(III)-containing ferritin, respectively (Supplementary Fig. 5). In the positive control using horse ferritin, Fe(III) is detected in the native state, and upon incubation with $Fe(II)Cl_2/H_2O_2$. However, the signal disappears when the ferritin is directly reduced by dithionite. In the case of *Mper*-mBfr, the Fe(III) is only detected when the protein is incubated with $Fe(II)Cl_2/H_2O_2$. Thus, the experiment confirms that *Mper*-mBfr exhibits the classical ferritin activity of Fe internalisation in vitro. The absence of Fe(III) in the as-isolated state might be due to the mild-reducing conditions during protein extraction and purification (e.g., anaerobic conditions and the presence of dithiothreitol and natural reducing agents from the microbial community).

## Phylogenetic analysis and classification of *Mper*-mBfr

To determine the phylogenetic placement of *Mper*-mBfr within the ferritin superfamily, we searched for homologous proteins (>30% identity and $<10^{-5}$ *e*-value) and constructed a phylogenetic tree of the 81 closest related protein sequences alongside experimentally validated ferritin family members (Fig. 7A). The resulting tree revealed that *Mper*-mBfr does not cluster within the established clades of ferritins, bacterioferritins, Dps or DpsL.

Since sequence identity alone did not provide insight into the quaternary structure of proteins in the phylogenetic tree, we explored whether protein structure prediction models could distinguish between 12- and 24-

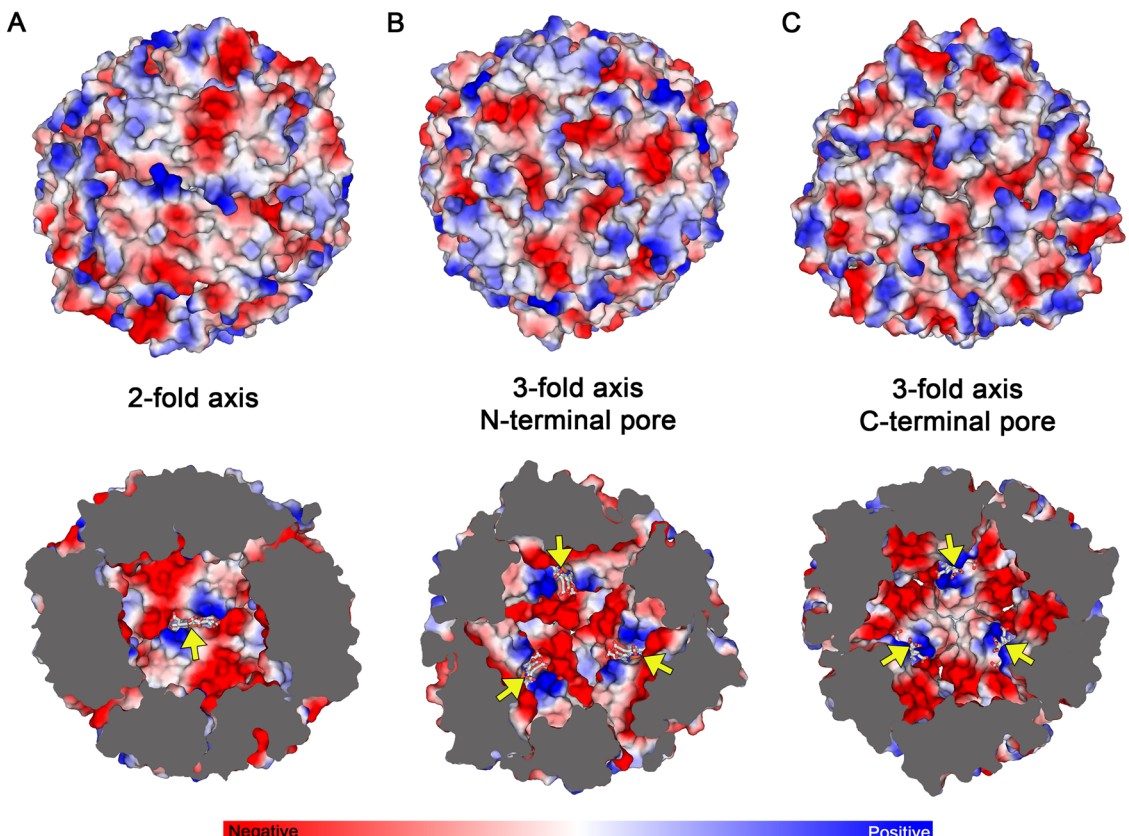

**Fig. 3 | Surface and inner cavity electrostatic charge profile of *Mper*-mBfr.** Surface representation of the outer (top) and inner (bottom) electrostatic charge profile along the 2-fold symmetry axis (**A**), 3-fold symmetry axis from the N-terminal pore view (**B**) and the 3-fold symmetry axis from the C-terminal pore view (**C**). Coprohemes are indicated by yellow arrows and displayed as sticks with their iron as spheres, with carbons, oxygen, nitrogen and iron coloured as white, red, dark blue and orange, respectively.

mer assemblies. Using ColabFold with an input of 12 copies of the amino acid sequence, we obtained structural predictions that formed either complete spheres or hemispheres (Supplementary data and Supplementary Fig. 6). To further validate this approach, AlphaFold3 was used with 24 copies of the amino acid sequences, which resulted in the prediction of either two 12-mers or a single 24-mer assembly. The predicted structures were additionally analysed for methionines positioned at homodimeric interfaces that are suitable for haem binding. These structural predictions provided a basis for classifying phylogenetically related sequences according to their quaternary structure, offering a complementary approach to sequence-based classification.

This method identified that out of the 81 closest related protein sequences, 71 proteins, across a wide range of microorganisms, were predicted to fold into 12-mer spheres while also encoding a methionine for haem binding. These proteins were classified as mini-bacterioferritins (mBfr).

Additionally, out of the 81 closest related proteins, seven were predicted to adopt a fold resembling a DpsL, while three aligned structurally with bacterioferritins. A comparison of the monomer structures of Dps, DpsL, Bfr and mBfr reveals that mBfr has the simplest composition, consisting solely of a four-helix bundle as its core structure. In contrast, other ferritin family proteins contain additional helices and flexible extensions (Fig. 7B). Dps and DpsL feature an additional helix positioned perpendicular to the four-helix bundle, located within the loop connecting helices B and C, at the position where (m)Bfrs bind haem. Some Dps and DpsL proteins, such as *Pa*-DpsL, possess additional N- or C-terminal tails associated with DNA-binding or endonuclease activity[21,38]. In ferritins and bacterioferritins, an additional C-terminal E-helix extends from the four-

helix bundle, forming the structural core of the four-fold symmetry axis in 24-subunit assemblies.

To identify sequence-level features that influence mBfr folding, we aligned the protein sequences of our predicted structures with the sequence of experimentally validated ferritin family proteins found in the RCSB PDB (Fig. 8A). The multiple sequence alignment of mBfr was then mapped onto a dimer structure using ConSurf[39]. The ferroxidase centre, consisting of Glu18, Glu51, His54, Glu93, Glu126 and His129 (numbering shifted +2 relative to the *Mper*-mBfr sequence), appears to be conserved among DpsL, mBfr and Bfr.

We then examined conserved residues involved in inter-subunit interactions. The haem-binding methionine is strictly conserved in mBfr, mostly present in Bfr, and absent in other ferritin family members. Beyond methionines, a combination of three highly conserved residues, His29, Glu56 and Glu60, is present at the dimer interface. In the mBfr crystal structure, these residues coordinate a cation (tentatively modelled as a sodium) that links the monomers together, with His29 showing particularly strong conservation among mBfr proteins. Another key pair of residues, Arg61 and Glu127, form a salt bridge stabilising dimer-dimer contacts. However, since these residues are also conserved in DpsL, Bfr and partially in Dps, they do not provide a distinguishing feature for mBfr.

Another interesting residue is Gly34, located in the loop between helices A and B, which appears to be conserved exclusively in 12-mers Dps, DpsL and mBfr. Taken together, the conservation of His29, Gly34, Met52 and an absence of N- and C-terminal extensions may serve as a useful signature for identifying more members of the mBfr group.

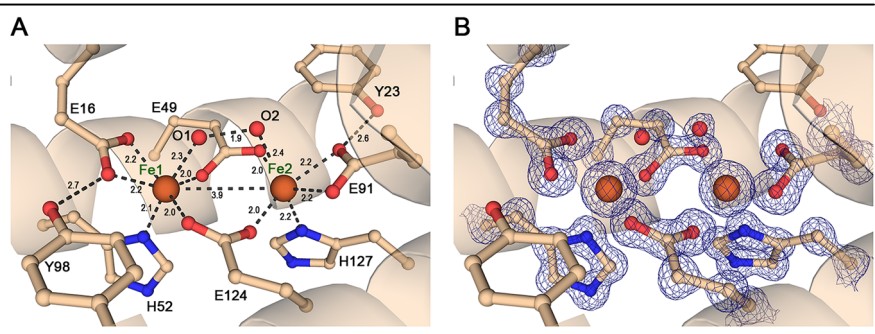

**Fig. 4 | Coproheme coordination in *Mper*-mBfr. A** Modelled coproheme with two alternative conformations and their overlap. The $2F_o-F_c$ map contoured at 1.0 σ is represented as a dark blue mesh. **B** Close-ups of the haem-binding pocket showing conformers independently. Beige and blue coils represent two different mBfr monomers. Oxygen, nitrogen and sulfur are coloured in red, dark blue and yellow, respectively. **C** Close-ups of the haem-binding pockets from the same view as (**B**) with a colour gradient reflecting the B-factors for each atom.

**Fig. 5 | Diiron ferroxidase centre of *Mper*-mBfr.**
**A** Close-up of the ferroxidase centre showing amino acid residues (balls and sticks) involved in iron coordination within the four-helix bundle in light orange cartoon. Distances are displayed in Å.
**B** Same view as (**A**) showing the amino acid residues involved in iron coordination, including the $2F_o-F_c$ map in dark blue contoured to 2.5 σ. Carbon, oxygen, nitrogen and iron are coloured as white, red, dark blue and orange, respectively. Waters O1 and O2 are in two alternative conformations.

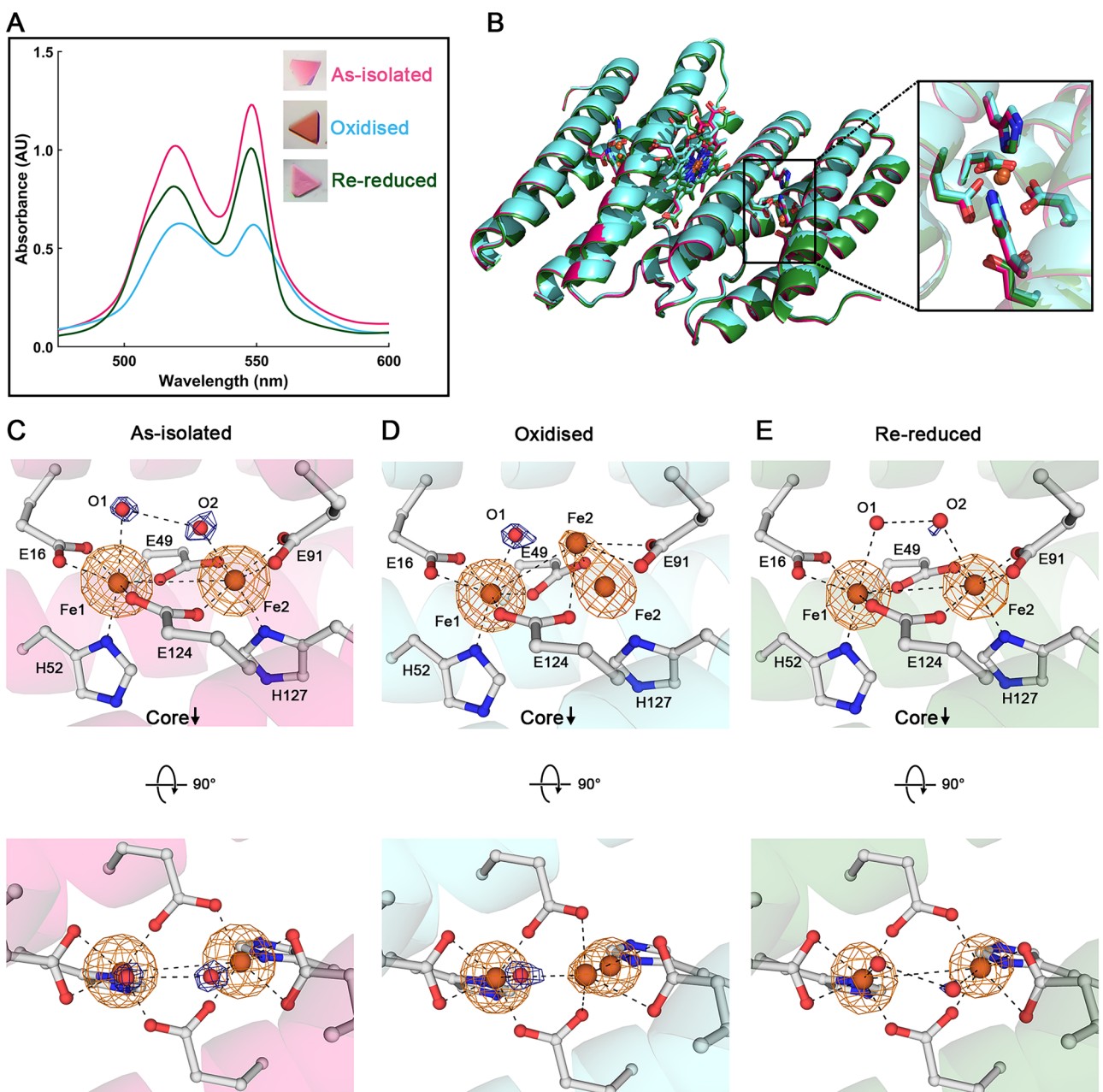

**Fig. 6 | Redox cycling of *Mper*-mBfr crystals. A** *In crystallo* spectra of the as-isolated crystal, partially oxidised upon 10 min $O_2$ exposure and oxidised upon 10 min $O_2$ exposure, then reduced by soaking the crystals for 4 min 40 s in 100 mM sodium dithionite. The inset displays photos of the corresponding crystals. **B** Overlay of a dimer of each structure, with as-isolated in pink, oxidised in cyan and re-reduced in dark green. The inset shows an overlay of the active site of all three conditions. **C–E** Zoom-in of the active site for all three conditions. Bottom panel displays a 90° rotation along the *x*-axis of the top panel. The dark blue and orange mesh represent the $2F_o$–$F_c$ map contoured to 2.5 σ for waters only, and the anomalous map contoured to 9 σ, respectively. Ligands are represented as balls and sticks, with carbon, oxygen, nitrogen and iron coloured in white, red, dark blue and orange, respectively. Dashes display close interaction between the Fe and their surroundings. In (**D**), interactions are only presented for Fe1 and the displaced Fe2.

To contextualise these findings, we compared the defining features of major ferritin subfamilies: classical ferritins, bacterioferritins, Dps, DpsL and the here-defined mBfr (Table 2). Among the family members, mBfrs stand out by their streamlined architecture, composed solely of a four-helix bundle, and their consistent 12-mer assembly with a strictly conserved haem-binding methionine. In contrast, other ferritin-like proteins show additional helices or tails associated with DNA binding. These distinctions emphasise the unique structural and functional identity of mBfrs within the broader ferritin superfamily and provide a framework for further classifying ferritin-like proteins.

## Discussion

Universally conserved across living organisms, ferritins have been shown to be more than an iron storage, with physiological functions that could be expanded to microbial stress response and DNA protection. Here, we expand our understanding of ferritin natural diversity by isolating and characterising mini-bacterioferritins, the first member of a previously unrecognised group within the ferritin family.

The assembly of *Mper*-mBfr into a 12-mer quaternary structure, combined with the incorporation of coprohaem in the same fashion as bacterioferritin, makes this ferritin-like protein unique. The only examples

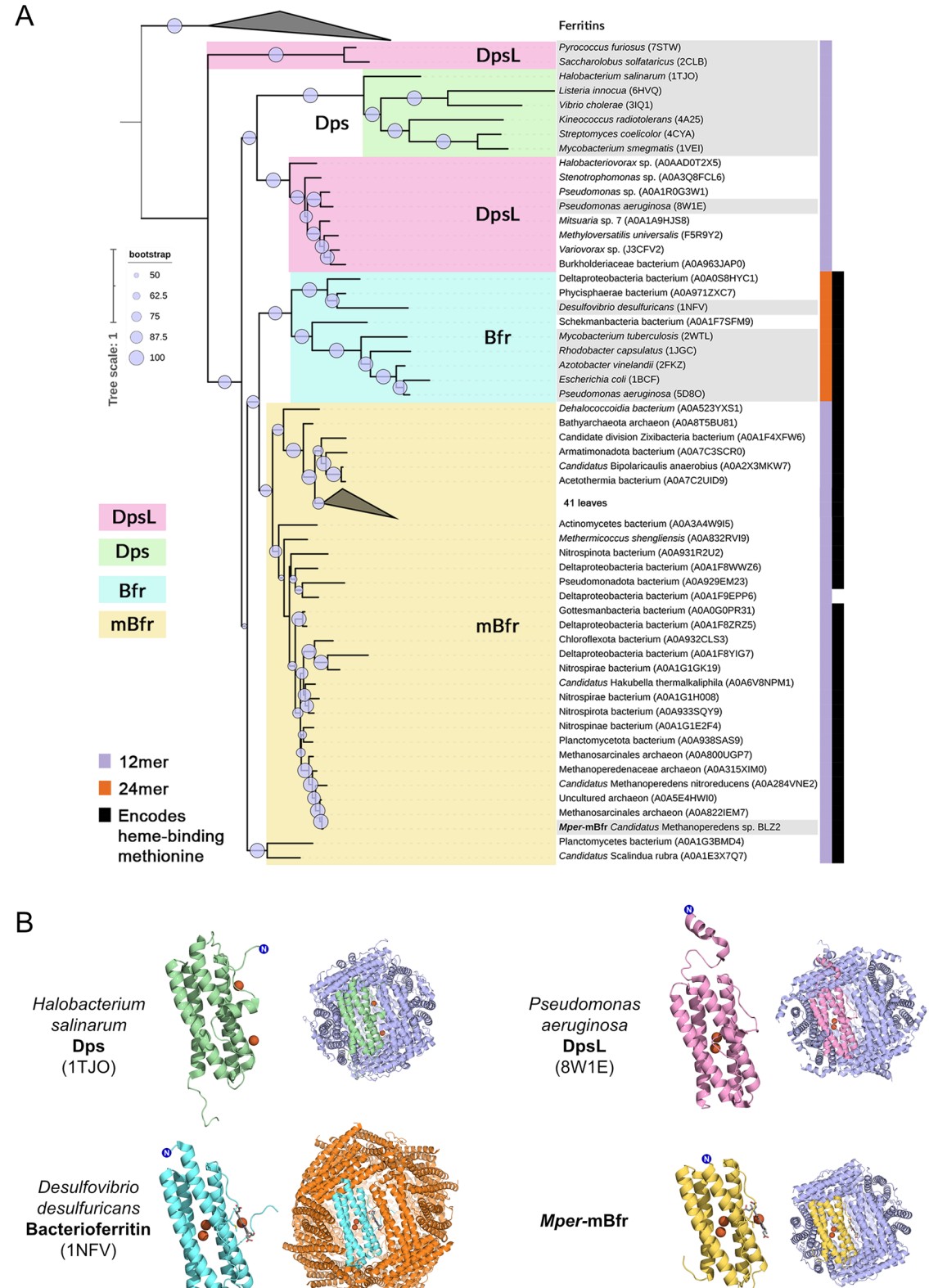

**Fig. 7 | Phylogeny of mini-bacterioferritins. A** Phylogenetic tree of amino acid sequences closely related to *Mper*-mBfr, together with sequences from solved protein structures (highlighted in grey). The clades in pink, green, cyan and yellow represent Dps-like proteins (DpsL), Dps, bacterioferritins (Bfr) and mini-bacterioferritins (mBfr), respectively. Multimerisation of the proteins was predicted with ColabFold and is indicated on the right as purple for 12-mers or orange for 24-mers. The presence of a haem-binding methionine is indicated in black. Bootstrap values in the range of 50–100 are indicated with purple circles. UniProt or RCSB PDB accession numbers are indicated in brackets. **B** Cartoon representation of four different types of ferritin-like proteins, including a monomer and its multimer. Colours of the monomers represent the different clades in the phylogenetic tree, and the colour of the additional subunits represents their multimer (purple for 12-mers and orange for 24-mers).

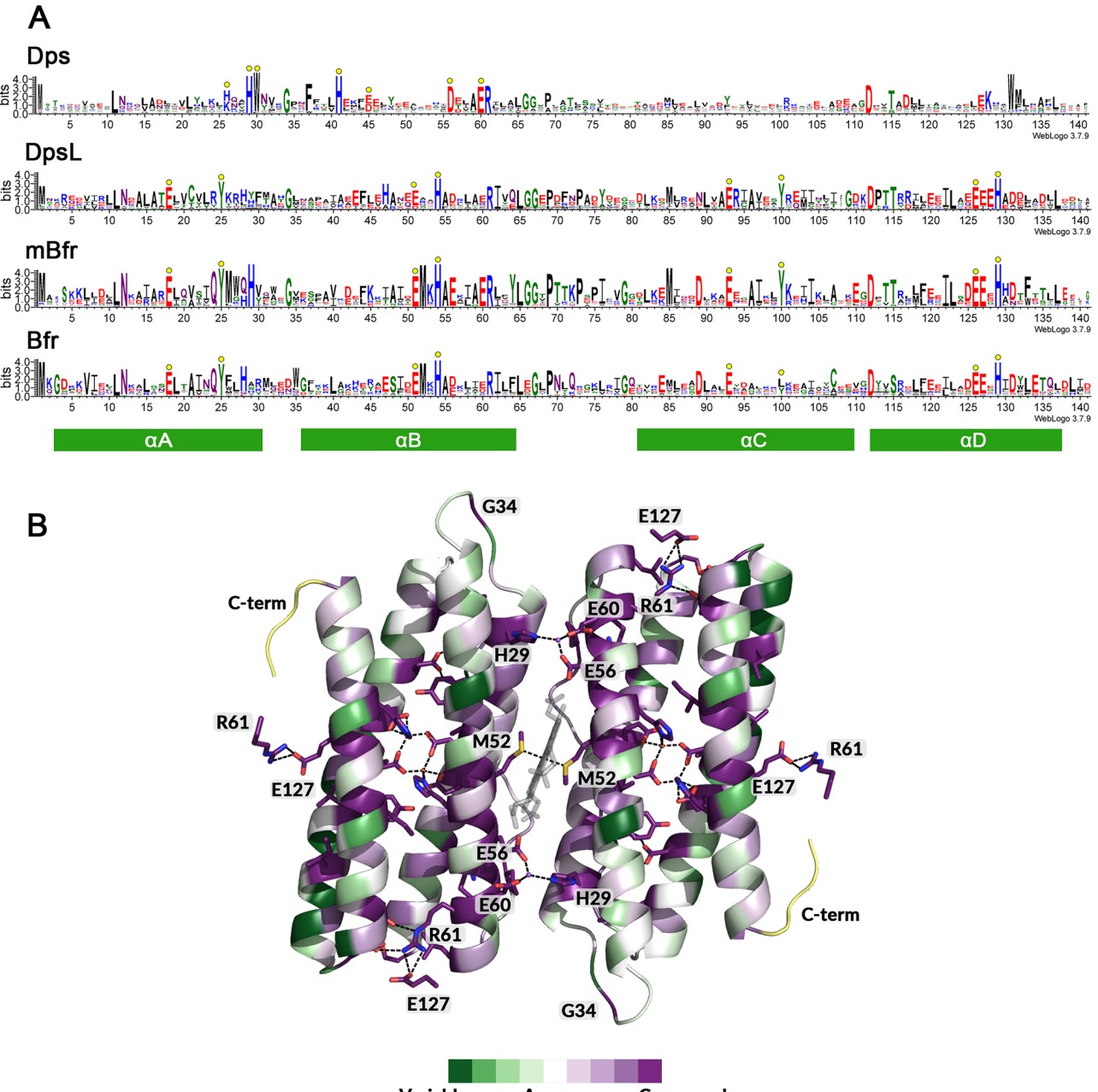

**Fig. 8 | Multiple sequence alignments of ferritin-family proteins. A** Multiple sequence alignments represented with WebLogo of Dps, Dps-like proteins (DpsL), mini-bacterioferritins (mBfr) and bacterioferritins (Bfr). Residues that are part of the active site are indicated by yellow circles. **B** ConSurf cartoon representation of the mBfr multiple sequence alignment showing conserved residues that interact with other amino acid chains. Regions with insufficient data are coloured yellow. Amino acid residues are numbered according to the multiple sequence alignment of mBfrs (+2 compared to *Mper*-mBfr). Ligands are represented as balls and sticks with oxygen, nitrogen and iron coloured in red, blue and orange, respectively.

of 12-mer ferritins binding a haem are Dps from *Porphyromonas gingivalis* (*Pg*-Dps) and DpsA from *Synechococcus elongatus* (*Se*-Dps). In *Pg*-Dps, the haem is bound in a completely different way via a single surface-exposed cysteine and likely serves as a mechanism for haem storage and protection against haem toxicity[40]. Regarding *Se*-Dps, the protein was found to bind haem *b* upon addition, though no specific binding mechanism was identified[41] and its sequence revealed no methionine residues positioned for haem coordination.

The incorporation of haem ligands facilitates the reduction of mineralised $Fe^{3+}$ in the protein core to soluble $Fe^{2+}$, as proposed in bacterioferritins[12]. This haem is typically a haem *b*[37]. However, in the case of

*Mper*-mBfr, the ligand was identified as a coprohem, an intermediate of the coproporphyrin-dependent (CPD) haem *b* biosynthesis pathway found in archaea[42,43]. Proteins that bind coprohem as a cofactor are rare and mainly limited to enzymes directly involved in the CPD pathway. Interestingly, the only other known example of a protein utilising coprohem as a cofactor is *Dd*-Bfr, which binds coprohem in a similar alternative dual conformation as *Mper*-mBfr[13,14]. Redox titration of *Dd*-Bfr revealed an unusually high redox potential of +140 mV, contrasting with other bacterioferritins that typically exhibit redox potentials below −200 mV[29]. This unique property raises intriguing questions about the function of coprohem in these proteins, particularly whether its presence influences electron transfer

**Table 2 | Overview of the different ferritin family members**

|  | Ferritin | Bacterioferritin | Dps | DpsL | Mini-bacterioferritin |
|---|---|---|---|---|---|
| Ecological distribution | Universal | Prokaryotes | Prokaryotes | Prokaryotes | Prokaryotes |
| Multimerisation | 24 | 24 | 12 | 12 | 12 |
| Haem incorporation | No | Yes, inter-subunit | No | No | Yes, inter-subunit |
| DNA binding | No | No | Yes, non-specific | Yes, non-specific (in most cases) | Unknown |
| Diiron site type | Canonical | Canonical | Inter subunit | Canonical | Canonical |
| Primary function | Iron storage & mobilisation | Iron storage and mobilisation | Stress response and DNA protection, limited iron storage | Stress response and DNA protection, iron storage | Unknown |

dynamics or interactions with physiological electron donors. The significantly higher redox potential suggests that coproheme may confer distinct reactivity compared to haem $b$, potentially affecting how these bacterioferritins participate in iron metabolism or oxidative stress responses. The reason for using coproheme instead of haem $b$ remains unclear but may reflect specific adaptations to anaerobic environmental conditions. The haem identity in other mini-bacterioferritins is presumably different from coproheme, as the Arg13 stabilising the two propionate groups can be substituted by an Ile (Fig. 8A).

A phylogenetic analysis revealed that *Mper*-mBfr is part of a larger group of proteins characterised by a 12-mer assembly that can coordinate a haem at the dimer interface. Except for two outliers, all proteins predicted to be mini-bacterioferritins cluster into two distinct subclades and are found in both bacteria and archaea. Previous studies suggest that the 12-mer DpsL from *Saccharolobus solfataricus* and *Pyrococcus furiosus* may resemble an ancient common ancestor from which 24-mer bacterioferritins evolved[3,5]. Although mini-bacterioferritins seem more closely related to bacterioferritins than to DpsL, the structure of the last common ancestor shared by these groups remains unclear. Our sequence and structural analyses support the hypothesis that mini-bacterioferritins retain an ancestral 12-mer configuration, which may have predated the divergence of ferritin family proteins, and subsequently acquired haem-binding properties. Alternatively, DpsL may have lost its haem-binding ability during evolution.

The structural transition between 12- and 24-mer remains an open question. One study tested whether the C-terminal helix E, present in bacterioferritins and ferritins but absent in Dps, plays a defining role in oligomerisation. When helix E of *E. coli* Bfr was fused to *Ec*-Dps, the quaternary structure increased in size, but helix E flipped outward, and the protein still assembled as a 12-mer[44]. In *Mycobacterium smegmatis*, a single Phe47Glu mutation was found to shift the oligomeric state to a 24-mer assembly, though only in the crystallised form and not in solution[45]. These findings suggest that while specific structural elements influence multimerisation, additional factors likely contribute to determining whether a ferritin-family protein assembles as a 12-mer or 24-mer.

The physiological function of *Mper*-mBfr remains speculative at this stage. Based on the Fe internalisation experiment (Supplementary Fig. 5), we concluded that the enzyme harbours ferritin activity in vitro[13] and might function as an iron-storage protein in vivo, similar to *Dd*-Bfr[13]. Additionally, it may contribute to oxygen protection, resembling the role of bacterioferritin in the anaerobe *Nitratidesulfovibrio vulgaris* (formerly *Desulfovibrio vulgaris*), which has been shown to mitigate oxidative stress[46]. A study on salt stress in '*Ca*. Methanoperedens sp.' Vercelli Strain 1 reported high expression of a 139-amino-acid ferritin-like protein with 87% sequence identity to *Mper*-mBfr, which protein modelling predicts to be a mini-bacterioferritin[47] (Supplementary Fig. 7). Interestingly, the gene encoding the protein was downregulated upon salt stress, suggesting a potential regulatory role in response to environmental changes.

The genome of '*Ca*. M. carboxydivorans' encodes four additional ferritin-like proteins, including two predicted rubrerythrins and two four-helix bundle proteins characteristic of the ferritin family (Supplementary Data, Supplementary Fig. 1). Based on the predicted tertiary structure of these two four-helix bundle ferritin-like proteins, it is difficult to assign them to any known ferritin subfamily, and modelling could not reliably predict their multimeric assembly. This genetic redundancy raises questions about the specific roles of these proteins and how they interact within the organism's iron metabolism, including the identity of the redox carriers involved in $Fe^{3+}$ reduction in the core, and the potential involvement of these proteins in stress responses such as protection against oxygen[46]. To fully understand the physiological role of *Mper*-mBfr, in vivo studies are now needed to investigate its regulation, function and potential interactions with other ferritin-like proteins in '*Ca*. Methanoperedens'. Pure culture, genetically editable microbes like *Methanosarcinales*[48] harbouring mini-bacterioferritin could represent an attractive alternative.

More broadly, this study investigated ferritins beyond well-studied model organisms, showcasing the rich and largely untapped structural diversity of ferritin-like proteins in nature. *Mper*-mBfr could represent an example of evolutionary diversity driven by niche-specific adaptations, where the incorporation of coproheme (a rare cofactor) and a 12-mer quaternary structure distinguishes it from other well-studied ferritins and bacterioferritins. By tapping into organisms that are challenging to cultivate and from non-isolated lineages, we provided a broader perspective on the ferritin family and their evolutionary path, highlighting the need to explore the natural microbial reservoir.

## Methods

### Bioreactor cultivation and lysis

Granular biomass highly enriched (20–40%) in '*Ca*. M. carboxydivorans' (formerly '*Ca*. Methanoperedens sp.' BLZ2) was obtained from a sequencing fed batch bioreactor as described previously[49]. About 30 g of anoxically stored biomass was defrosted in lukewarm water. Under a $N_2/CO_2$ (90:10) atmosphere, biomass was resuspended in a total volume of 90 mL IEC A buffer (50 mM Tris/HCl pH 8.0, 2 mM dithiothreitol (DTT)). The suspension was sonicated (30 s, 75% power, 4 cycles, 1 min breaks, SONO-PULSE Bandelin) to disrupt the granules, followed by five rounds of French Press at approximately 1,000 PSI (6.895 MPa). To prevent oxygen contamination, the French Press cell was flushed with $N_2$ and washed twice with anoxic IEC A buffer. Cell debris were removed by centrifugation (45,000 × g, 30 min at 18 °C). Supernatant was sonicated again (30 s, 75% power, 3 cycles, 1 min breaks) to reduce viscosity and subsequently subjected to ultracentrifugation (100,000 × g, 1.5 h at 4 °C, rotor 70.1Ti, Beckman Coulter) to prepare a soluble protein extract.

### Mini-bacterioferritin purification

Proteins were purified under anoxic conditions in an anaerobic tent containing a $N_2/H_2$ (97:3) atmosphere at 20 °C under yellow light. The soluble extract (90 mL, 7.8 mg mL$^{-1}$) was divided into two and loaded directly on 2 × 5 mL HiTrap Q-Sepharose HP columns (GE, Healthcare, Munich, Germany) pre-equilibrated with IEC A buffer. The loaded column was washed with IEC A buffer, and proteins were eluted with a 0–65% gradient of IEC B buffer (50 mM Tris/HCl, pH 8.0, 1 M NaCl, and 2 mM DTT) in 80 min at 2 mL min$^{-1}$ flow rate. Two millilitre fractions were collected over the elution. Proteins were tracked using multi-wavelength absorbance

monitoring ($\lambda$ 280, 424 and 550 nm) in combination with denaturing PAGE. Fractions of interest were pooled and diluted 1:1 with HIC B buffer (25 mM Tris/HCl pH 8.0, 3 M $(NH_4)_2SO_4$, and 2 mM DTT), filtered (0.2 $\mu$m nitrocellulose, Sartorius, Germany), and loaded on a 1.622 mL Source 15PHE 4.6/100 PE column (GE Healthcare, Munich, Germany) pre-equilibrated with HIC B buffer. Proteins were eluted using a 66 to 0% gradient of HIC B buffer mixed with HIC A buffer (25 mM Tris/HCl pH 8.0, and 2 mM DTT) in 70 min at 0.7 mL min$^{-1}$ and collected in 0.5 mL fractions. Fractions of interest were pooled again and concentrated using a centrifugation concentrator (10 kDa cut-off, Vivaspin, Sigma-Aldrich, Germany). The final purification step included size exclusion chromatography on a Superdex 200 Increase 10/300 GL (GE Healthcare, Munich, Germany) in 25 mM Tris/HCl pH 8.0, 10% v/v glycerol, and 2 mM DTT with a flow rate of 0.4 mL min$^{-1}$, where *Mper*-mBfr showed an elution volume of 11.2 mL. This last step was skipped for the purification that yielded crystals for the redox cycle experiment. The final fractions of interest were concentrated and directly used for crystallisation or stored anoxically under N$_2$ atmosphere at $-80$ °C.

### Protein identification by mass spectrometry

*Mper*-mBfr was identified using matrix-assisted laser desorption ionisation time-of-flight mass spectrometry (MALDI-TOF-MS) as previously described[50]. A spectrum range of 450–3000 *m/z* was recorded on Microflex LRF MALDI-TOF (Bruker, Billerica, MA, USA). Spectra were analysed using BioTools software (version 3.2, Bruker Life Sciences) linked to MASCOT search engine (Matrix Science Ltd., London, UK) loaded with the proteome of '*Ca.* M. carboxydivorans' (NCBI:txid2035255[24]). Peaks were selected with a minimal mass difference of 0.3 Da, a signal-to-noise threshold of 3, and a quality factor threshold of 20%. MASCOT search parameters included one trypsin miscleavage, global modifications of carbamidomethylated cysteines, variable modification of oxidised methionines and a mass deviation of 0.2 Da. Eight observed peaks correlated with predicted *Mper*-mBfr peptides, resulting in a MOWSE score of 70.

### Haem identification by mass spectrometry

Haem was extracted from *Mper*-mBfr using a method from ref. 51. To obtain enough haems, several protein solutions (20–100 $\mu$L, 5–42 $\mu$g mL$^{-1}$) were combined. One millilitre acetonitrile:HCl (ACN 0.8:0.2 HCl 1.7 M) was added to up to 50 $\mu$L protein solution and incubated on a shaker at room temperature for 20 min. Then, 250 $\mu$L saturated MgSO$_4$ and 25 mg NaCl were added, and the solution was incubated on a shaker at room temperature for 5 min. This solution was then centrifuged (2500 × *g* for 5 min) to obtain phase separation. The organic top layer containing the haems was then extracted and stored at 4 °C.

The extract was analysed using an Agilent 1290 Infinity II LC system coupled to a 6546 Quadrupole Time of Flight mass spectrometer. A volume of 0.5 $\mu$L of the haem extract was injected onto a Poroshell 120 column (EC-C18, 2.1 × 50 mm, 1.9 $\mu$m; Agilent) maintained at 25 °C, followed by elution with a gradient of mobile phase A (0.2% formic acid in water) and B (0.2% formic acid in acetonitrile) at a flow of 0.3 mL min$^{-1}$. The gradient was as follows: 0–15 min: 10–100% B, 15–25 min: 100% B, 25–26 min: 100–10% B, followed by 3 min re-equilibration at 10% B. One minute after injection, the eluate was directed to the Q-ToF MS operated in the positive ionisation mode. MS1 scans were collected for *m/z* 50–1200, at a scan rate of 4 spectra per second. MS2 fragmentation spectra were collected using the same setup, but collecting targeted MS2 scans of *m/z* 706.2 at collision energies of 20, 40 and 60. MS1 and MS2 spectra of the peak with *m/z* 706.2 were imported into SIRIUS v6.10[52,53] for the prediction of elemental composition and structure. The elemental composition was determined using de novo prediction, limiting the number of nitrogens to 4 based on the porphyrin structure observed in the crystal structure.

### High-resolution clear native PAGE (hrCN PAGE)

The PAGE was prepared and run as previously[54]. The whole process was performed anoxically. Glycerol (20% v/v final) was added to each sample, and 0.001% (w/v) Ponceau S serves as a marker for protein migration. The electrophoresis cathode buffer contained 50 mM Tricine; 15 mM Bis-Tris, pH 7; 0.05% (w/v) sodium deoxycholate; 0.01% (w/v) dodecyl maltoside and 2 mM DTT. The anode buffer contained 50 mM Bis-Tris buffer pH 7.0, and 2 mM DTT. hrCN PAGE was carried out using a 5 to 15% linear polyacrylamide gradient, and gels were run with a constant 40 mA current (PowerPac™ Basic Power Supply, Bio-Rad).

### Ferritin activity and specific staining

*Mper*-mBfr was purified using a simplified protocol involving ammonium sulfate precipitation, with all steps performed under anaerobic conditions in a Coy tent containing a N$_2$/H$_2$ atmosphere at a 97:3 ratio. 15.3 g of wet-weight cells were diluted in 15 mL of lysis buffer as described above. Cells were lysed by 10 cycles of sonication (for 10 s followed by 20 s break, at 70% amplitude). The lysate was anaerobically centrifuged at 45,000 × *g* for 30 min at 20 °C. The supernatant was kept anaerobically, and the pellet was resuspended in 25 mL lysis buffer for another round of sonication (10 cycles with 10 s followed by 20 s break, at 95% amplitude). The second lysate was centrifuged one more time at 45,000 × *g* for 30 min, 20 °C. Both supernatants were combined and centrifuged anaerobically at 180,000 × *g* for 45 min. 33.94 g of (NH$_4$)$_2$SO$_4$ were added to the 72 mL of supernatant to reach a 70% saturation. The 30-min incubation at room temperature under mild agitation led to heavy aggregation. The sample was centrifuged for 30 min at 6000 × *g* at 20 °C, and the supernatant was passed through a 0.2 $\mu$m filter (Sartorius, Germany). The filtrate was applied to a 5 mL Phenyl-Sepharose HP (GE, Healthcare, Munich, Germany) pre-equilibrated with HIC B buffer. After washing with HIC B buffer, the proteins were eluted with a 3.0–0.6 M (NH$_4$)$_2$SO$_4$ linear gradient for 60 min at 2 mL min$^{-1}$. The fraction containing *Mper*-mBfr was pooled and concentrated through a 30-kDa cutoff centricon. During ultrafiltration, the buffer was exchanged for 25 mM Tris/HCl, pH 8.0, 10% v/v glycerol, and 2 mM DTT. The final protein preparation containing 12.1 mg mL$^{-1}$ of proteins in 77 $\mu$L was aliquoted, frozen anaerobically by adding an additional 0.6 bars of N$_2$, and stored at $-80$ °C. This protein enrichment method led to a preparation containing two contaminants, as witnessed in Supplementary Fig. 5 by the two bands comprised between 20 and 66 kDa.

To test the ferritin activity, *Mper*-mBfr was submitted to three different treatments: (i) incubation at 1 mg mL$^{-1}$ in a 100 mM potassium phosphate buffer pH 7.0 for 45 min, (ii) incubation at 1 mg mL$^{-1}$ in a 100 mM potassium phosphate buffer pH 7.0, 5 mM sodium dithionite for 45 min and (iii) incubation at 1 mg mL$^{-1}$ in a 100 mM potassium phosphate buffer pH 7.0, 2 mM Fe(II)Cl$_2$, 0.2 mM sodium dithionite, followed by three sequential injections of H$_2$O$_2$ (equal amounts of H$_2$O$_2$ added to reach a final 2 mM concentration at the end of the incubation), each time a 15-min incubation, for a total of 45 min incubation. All experiments were performed at 20 °C in 0.2 mL tubes containing a final reaction mixture volume of 30 $\mu$L, in an anaerobic chamber filled with a N$_2$/H$_2$ (97:3 ) atmosphere. After incubation, 10 $\mu$L of a solution of 80% (v/v) glycerol and 0.004% (w/v) Ponceau S was added to the samples. hrCN PAGE was run aerobically using 5–15% acrylamide gradient gels. 4.5 $\mu$g of protein were loaded. After migration, the gel was cut into three parts. The first part was stained with protein staining solution for 12 h. Iron staining was performed with Ferene S (1 mM Ferene S, 2% (v/v) acetic acid, 0.1% (v/v) thioglycolic acid) for 30 min at room temperature. The detection of Fe(III) was done by Prussian Blue staining (2% (w/v) potassium hexacyanoferrate(II) trihydrate, 2% (v/v) HCl) for 30 min at room temperature. As a control, the ferritin from equine spleen type I (Sigma-Aldrich, Germany, reference F4503-25MG) was submitted to the same treatment, except that the protein concentration in the three treatments was 0.5 mg mL$^{-1}$.

### Crystallisation

An initial screening using the sitting drop method was performed on a 96-well MRC 2-well polystyrene crystallisation plate (SWISSCI, Switzerland) at 20 °C under a pure N$_2$ atmosphere. The plate was filled with 90 $\mu$L crystallisation solution (Wizard screen from Jena Bioscience, Germany) in the

large reservoir, and the purified protein solution concentrated to 4.4 mg mL$^{-1}$ was spotted in the small reservoirs by mixing 0.5 µL purified protein with 0.5 µL crystallisation solution using an OryxNano robot (Douglas Instruments Ltd, UK). Once prepared, the plate was transferred to an anaerobic chamber containing a $N_2/H_2$ (97:3) atmosphere for long-term storage. The crystal diffracting to atomic resolution was harvested from the original screening plate containing the following crystallisation solution: 20% v/v Jeffamine M-600, pH 7.0, and 100 mM HEPES, pH 7.5. Prior to liquid nitrogen freezing, the crystal was soaked in the crystallisation solution supplemented with 25% v/v glycerol for a few seconds.

For the redox cycling experiment, crystals were obtained on a Junior Clover plate (Jena Bioscience, Germany) using 100 µL crystallisation solution in the reservoir composed of 20% w/v Polyethylene glycol 3,350 and 200 mM tri-Potassium citrate. One microlitre of purified protein solution at 22 mg mL$^{-1}$ was mixed with 1 µL of the crystallisation solution. The crystal corresponding to the as-isolated state was directly harvested inside the anaerobic chamber, and the one corresponding to the oxidised state was exposed to ambient air for 10 min. For the re-reduced state, the crystal was exposed to ambient air for 10 min, then soaked in the crystallisation solution supplemented with 100 mM of freshly prepared sodium dithionite. All harvested crystals were soaked for a few seconds in the crystallisation solution supplemented with 20% v/v glycerol, except for the re-reduced state, where the crystal was soaked in the crystallisation solution supplemented with 20% v/v glycerol and 90 mM sodium dithionite.

### X-ray collection and model refinement/validation

Data were collected (see Table 1) at the Swiss Light Source (SLS, beamline PXI-X06SA) and at the European Synchrotron Radiation Facility (ESRF, beamline BM07-FIP2) under cryogenic conditions (100 K). All data were integrated with *autoPROC*[55]. All datasets were treated as isotropic, except for the atomic resolution dataset, which exhibited a slight anisotropy. The atomic resolution structure was solved by molecular replacement using *Phaser* from the *PHENIX* package[56] with an Alphafold2 model[57] of the *Mper*-mBfr monomer. The three other structures from the redox cycle were solved in the same way, with the atomic resolution model as a template.

All models were manually optimised with *Coot* (v0.9.8)[58]. Refinement was performed with *Phenix* refine (v1.21.1-5286)[56] without applying non-crystallography symmetry. All models were refined by using a translation-libration screw, except for the atomic resolution model, in which all atoms were considered anisotropic. All models were refined by adding hydrogens in riding positions. Models only contain hydrogens on the protein chains and not on ligands and solvent. The final structures deposited to the PDB did not contain hydrogens except for the atomic resolution dataset. The different structures were validated by the *MolProbity* tool integrated with *PHENIX*[59].

### Structural analyses

The inner volume of *Mper*-mBfr, *Hs*-Dps, *Pa*-DpsL and *Dd*-Bfr was measured using the programme HOLLOW[60], by modelling a cylinder in the inner cavity using a grid spacing of 0.5 Å in the case of *Mper*-mBfr, *Hs*-Dps and *Pa*-DpsL or 1.0 Å in the case of *Dd*-Bfr and manual curation in PyMOL (v2.2.0, Schrödinger, LLC). PyMOL was further used for structural analysis and visualisation of protein structures.

### UV-Vis absorption spectroscopy

UV-Vis spectra were recorded at a protein concentration of 2 mg mL$^{-1}$ on a Cary 60 (Agilent Technologies, USA) in a stoppered anoxic quartz cuvette with a light pathlength of 1 cm. The oxidised spectrum was recorded after opening the stoppered cuvette and equilibrating with air for 10 min. The same sample was reduced again inside an anaerobic chamber with a $N_2/H_2$ (97:3) atmosphere by adding freshly prepared sodium dithionite (100 mM stock solution) to a final concentration of 1 mM, and stoppering the cuvette. The spectrum was recorded after 5 min.

UV-Vis absorption spectra of *Mper*-mBfr crystals were collected at the *in crystallo* Optical Spectroscopy (*ic*OS) laboratory at the ESRF[61]. Spectra

were measured using a DH-2000-BAL deuterium-halogen lamp (Ocean Optics, Dunedin, FL) as the reference light source and a QE65 Pro spectrophotometer (Ocean Optics, Dunedin, FL).

Crystals were mounted in a standard crystal-mount loop between the two reflective objectives of the *ic*OS setup and maintained at 100 K using an evaporating nitrogen cryostream (Oxford Cryosystems). The white lamp was connected to the upper objective via a 200 µm-diameter fibre. The spectrophotometer was connected to the lower objective via a 400 µm-diameter fibre. Spectra were recorded with a 6 ms acquisition time and averaged over 10 scans. Spectra were prepared using the *ic*OS toolbox[62] with baseline correction performed using the absorbance in the 780–800 nm range and scaled to each other based on the absorbance at 400 nm (Supplementary Fig. 3). Additional scattering corrections were applied to the [450–600 nm] region in order to better compare the peaks of interest (Fig. 6A).

### Phylogenetic analysis

The ferritins amino acid sequences used to build the phylogenetic analyses in Fig. 7 were recovered from a database constructed by combining all sequences associated with InterPro IPR002024 (Bacterioferritin) and Pfam PF00210 (Ferritin-like domain) protein families. Duplicated sequences were removed using seqkit v.2.8.0 (function rmdup)[63] resulting in a total of 68,990 unique protein sequences in the database. A DIAMOND v.2.1.6[64] blastp search was done using the sequence of *Mper*-mBfr (UniProt A0A0P8A5A1) as reference. The top 1000 matches that respected a cut off of >30% identity and <10$^{-5}$ *e*-value were used in the final analysis. Additionally, 15 ferritin family protein sequences with experimentally confirmed structures were obtained from the RCSB PDB. UniProt or RCSB PDB accession numbers of the proteins used in the alignment are displayed in brackets in Fig. 7. The sequences were aligned using MAFFT v7.397[65], and regions in the alignment containing >50% gaps were removed using the web version of ClipKIT[66]. The tree was constructed using IQ-TREE v2.3.6[67], where the best model (LG + F + I + G4) was chosen according to the Bayesian Information Criterion (BIC), and 1000 bootstraps were executed. The final visualisation and annotation of the phylogeny were done in iToL v7.1[68].

To predict the multimerisation of the proteins in the tree, ColabFold[69] was used, giving 12 amino acid chains input per protein, using 1-5 models and 3 recycles per protein. The predicted models were analysed in PyMOL (v2.2.0, Schrödinger, LLC), showing multimers folded in either a perfect 12-mer sphere or a 12-mer hemisphere, indicating a likely 24-mer multimerisation. Examples of the predicted structures are shown in Supplementary Fig. 6, and the confidence matrices for all predicted structures are provided in Supplementary Data.

To visualise the conserved residues, the Consurf server[39] was used with the atomic resolution *Mper*-mBfr structure together with the multiple-sequence alignment of all predicted mBfrs (Supplementary Data) and otherwise default parameters.

### Statistics and reproducibility

No statistical methods were applied beyond those implemented within the software tools described in the 'Methods' section. Crystallisation of *Mper*-mBfr was reproducible, and high-resolution X-ray diffraction data were obtained on independent occasions.

### Use of generative artificial intelligence

The authors used AlphaFold2, -3 and ColabFold for protein structure prediction as described in the Methods. In addition, ChatGPT (OpenAI) was used solely to assist with language refinement during manuscript preparation. No AI tools were used to generate or alter experimental data or scientific conclusions. All AI-assisted content was reviewed and approved by the authors, who take full responsibility for the final manuscript.

### Data availability

All protein models and their associated structure factors were deposited in the Protein Data Bank with PDB accession codes: 9QQ4 (*Mper*-mBfr as-

isolated), 9QQI (*Mper*-mBfr without treatment), 9QQ5 (*Mper*-mBfr O₂ exposed), 9QQH (*Mper*-mBfr O₂ exposed then DT reduced). Source data for the graphs in Figs. 1B, C, 6A, 7 and 8 can be found in Supplementary Data. X-ray raw data will be made available on request.

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

## Acknowledgements

We would like to thank the Max Planck Institute for Marine Microbiology and the Max Planck Society for their continuous support. We thank Christina Probian and Ramona Appel of the Microbial Metabolism laboratory for their assistance and support, and Marie-Caroline Müller for performing the initial crystallisation screen. We thank Huub op den Camp for his help with protein mass spectrometry. We acknowledge the SLS and ESRF synchrotrons for allocating beam time and the beamline staff of PXI-X06SA and BM07-FIP-2 (Thanks to the proposal number 20230522) for their advice and assistance during data collection. T.W. was supported by a European Research Council consolidator grant [Grant number 101125699] and the Max Planck Society. The initial crystallisation screening performed by an OryxNano robot was financed by the DFG priority programme 1927 Iron-Sulphur for Life [Grant number WA 4053/1-1]. This study was supported by the SIAM Gravitation grant funded by NWO [Grant number 024.002.002] and an NWO-VIDI Talent grant [Grant number VI.Vidi.223.012]. It was furthermore supported by the ERC Synergy Grant MARIX [Grant number 854088]. We acknowledge the French Biology/Health Panel Review Committee for the provision of synchrotron radiation beamtime at the ESRF (Grenoble, France) on beamline BM07-FIP2, supported by the French ANR PIA3 (France 2030) EquipEx+ project MAGNIFIX under grant agreement ANR-21-ESRE-0011. This work used the *ic*OS platform of the Grenoble Instruct-ERIC centre (ISBG; UAR 3518 CNRS-CEA-UGA-EMBL) within the Grenoble Partnership for Structural Biology (PSB), supported by FRISBI (ANR-10-INBS- 0005-02) and GRAL, financed within the University Grenoble Alpes graduate school (Ecoles Universitaires de Recherche) CBH-EUR-GS (ANR-17-EURE-0003). The ESRF is acknowledged for access to the *ic*OS Laboratory via its in-house research program.

## Author contributions

Microbial cultivation was done by M.W. and C.U.W. Purification and crystallisation were performed by M.W. and T.W. Ferritin Fe-incorporation assay was performed by M.B. and O.N.L. X-ray data collection was done by S.E. and T.W. X-ray data processing, model refinement and analysis were done by M.W. and T.W. *In crystallo* UV-Vis absorption spectroscopy and analysis were done by S.E. and A.R. Protein mass spectrometry was done by M.W., and haem mass spectrometry was done by R.S.J. Phylogeny analyses were performed by M.W. and P.L. Funding for this work was obtained by M.S.M.J, C.U.W., A.R. and T.W. The manuscript was written by M.W. Manuscript edition and corrections were made by all authors.

## Funding

## Competing interests

The authors declare no competing interests.
