## [Transparent Peer Review file · Communications Biology]

Mini-bacterioferritins: Structural insight into a ferritin-like protein from the anaerobic methane-oxidising archaeon *Candidatus Methanoperedens carboxydivorans*

Corresponding Author: Dr Tristan Wagner

Version 0:

Reviewer comments:

Reviewer #1

(Remarks to the Author)

This manuscript reports on the structure and biochemical characterization of a novel member of the ferritin family, which has been named “mini-bacterioferritin” (Mper-mBfr). The authors isolated the mini-bacterioferritin from an enriched culture of the methanotrophic archaeon *Candidatus Methanoperedens* BLZ2. The crystal structure of the novel protein revealed a 12-mer protein assembled from identical subunits consisting of a 4-helix bundle that lacks the E-helix characteristic of bacterioferritins and the N' or C'-terminal extensions found in the subunits of Dps and Dps-like (DpsL) proteins. Unlike the 12-mer Dps, where the ferroxidase ligands are provided by two subunits in a dimer, the ferroxidase ligands in Mper-Bfr are contained within each subunit and are identical in composition to the ferroxidase ligands in bacterioferritin. Although these characteristics would suggest a 12-mer DpsL, the Mper-mBfr harbors a conserved set of Met residues (M50 in Mper-mBfr) which function as axial ligands to coproheme, which are attributes not found in DpsL. Taken together, the evidence supports a novel member of the ferritin family.

The manuscript articulates the findings clearly. A few issues should be addressed before the manuscript can be accepted for publication.

1) The discussion indicates that Mper-mBfr probably serves as an iron storage protein, but the manuscript presents no evidence supporting this highly relevant idea. If Mpr-mBfr indeed stores iron in its interior cavity, it is probable that the protein was isolated with an iron core. This can be readily determined in at least two ways, as has been shown for natively isolated Bfr from *Pseudomonas aeruginosa* and from *Acinetobacter baumannii*.

a) The UV-vis spectrum of Mper-mBfr encompassing the wavelength range 280-800 nm should be included in the manuscript. The presence of iron in the core of Mper-mBfr should give rise to an intense feature near 300 nm.

b) The presence of iron in the Mper-mBfr core can also be visualized in a native PAGE gel stained with Ferene.

c) If an iron core is not observed in the as isolated protein, attempts to reconstitute an iron core should be carried out in vitro, followed by analysis of the iron content to determine whether iron can be accumulated inside Mper-mBfr.

2) Figure 6 and the accompanying text indicate that the di-iron ferroxidase center is in the ferrous oxidation state. Exposure to oxygen leads to loss in electron density from site Fe2, with new electron density developing at a new site, also labeled Fe2. The electron density attributed to iron in the structures should be corroborated by acquiring diffraction data near iron's absorption edge to calculate anomalous difference maps.

3) Is the new Fe2 site observed upon oxidation closer to the protein interior, or to the protein exterior? Figure 6 should be modified to provide this context. The former would suggest that oxidation of the di-iron center leads to internalization of Fe(III).

4) Exposure of the crystals to oxygen for 10 minutes affects the Fe2 site but appears to leave the Fe1 site intact. The authors should explore longer exposure to oxygen to investigate the fate of Fe1. Will both iron ions in the di-iron center be lost from

the ferroxidase center upon oxidation? This could indicate that the ferroxidase center in Mper-mBfr may act as a catalytic site and pore for Fe(III) internalization, as is the case in Bfr from *P. aeruginosa*.

Reviewer #2

(Remarks to the Author)

In this work the authors describe a new class of ferritin like proteins. The overall findings and structural work presented is very exciting and of significant interest to the field. I'm excited to see this work published and only have a few minor points that should be addressed in the manuscript.

-In the introduction there is a description of the difference types of ferritin like biomolecules. However, there is no mention of the difference between heavy and light chain ferritins in the standard ferritins in the introduction. For example: "Ferritins oligomerise as 24-mer sphere-like structures (2,3,7–9). Each subunit contains a nuclear diiron ferroxidase centre that oxidises Fe²⁺ to Fe³⁺ using either oxygen or hydrogen peroxide (10). The iron is then mineralised and stored inside the hollow spherical cavity of the protein, which can hold up to 4,500 iron atoms (11)". Here it would be good to mention that the heavy chain contains the ferroxidase center while the light chain has a site for mineral nucleation. While I don't think it is important to this work to go too deep in to this, it should at least be mentioned for newcomers to the field of ferritin structure-function.

-In Figure 6D,E,F, it would be helpful to see this same view but with a lower contour to see the strength of the density around the oxygens. This could be shown in a supplemental figure.

-Regarding the coproheme discussed here: "The atomic resolution electron density map revealed a dual conformation of the coproheme ligand, with each orientation present at approximately 50% occupancy (Fig. 4A)" Was the occupancy refined during refinement of the model? When you say approximately 50%, was the model refined to 50%? I think it would be helpful to give the value it actually refined to instead of just saying 50%. If what is meant by the "approximately 50%" is that I wasn't completely clear, maybe mention that with a bit more detail in the text. With this quality of structure I would think you would be able to provide a more clear number, but maybe that isn't the case. either way, some more detail here would be helpful.

Version 1:

Reviewer comments:

Reviewer #1

(Remarks to the Author)

The authors have addressed most of the concerns raised in the original review. There is, however, the remaining issue of whether the novel ferritin-like protein (Mper-mBfr) can store iron in its interior cavity. This concern had been brought up in the previous review but was not properly addressed in the revised manuscript. This important issue, which is aimed at determining whether the novel Mper-mBfr protein exhibits the classical iron storage function must be properly dealt with before the manuscript can be accepted for publication. More detailed explanation and some suggestions are outlined below.

The UV-vis spectrum in Figure 1 of the revised manuscript shows a small peak at 310 nm, which prompted the authors to speculate "might indicate the presence of Fe(III) (Fig. 1B)". It is highly improbable that this peak originates from Fe(III)-O- moieties in the interior cavity; the presence of Fe(III)-O- gives rise to a broad, intense band which does not peak at 310 nm. Furthermore, the small 310 nm peak disappears in the presence of oxygen, providing further support for the idea that the small peak does not originate from Fe(III)-O-. Rather, the small 310 nm peak is probably a feature of the Fe-porphyrin system. Consequently, the spectra in Fig. 1 indicate that the protein, as isolated, is devoid of an iron core, and the authors should indicate this in the results and discussion.

In the revised manuscript the authors made no attempt to reconstitute an iron core in vitro, despite the concern having been raised in the previous review (1.c.): "If an iron core is not observed in the as isolated protein, attempts to reconstitute an iron core should be carried out in vitro, followed by analysis of the iron content to determine whether iron can be accumulated inside Mper-mBfr".

Reconstitution of an iron core in vitro is an important and commonly performed biochemical assay carried out with ferritin and ferritin-like proteins to determine whether the ferroxidase centers are functional and if the protein accumulates iron in its interior. Consequently, attempts to reconstitute an iron core in the novel Mper-mBfr should be performed and the results reported in the manuscript. This assay can be carried out anaerobically in a titration where the addition of each Fe(II) aliquot is followed by the addition of an equivalent of H₂O₂ (e.g. ref. 21). Reconstitution of an iron core would indicate that one of the functions of the novel Mper-mBfr is in the detoxication of Fe(II) and storage of Fe(III) (the classical iron storage function) which is one role attributed to most Dps and Dps-like proteins. On the other hand, if an iron core cannot be reconstituted in vitro, together with the fact that the isolated protein is devoid of an iron core, it would suggest an entirely different function, such as a possible role in the mitigation from H₂O₂-mediated stress.

Reviewer #2

(Remarks to the Author)

The authors have addressed my small concerns in the initial submission. I would support publications of this work.

Version 2:

Reviewer comments:

Reviewer #1

(Remarks to the Author)

The authors have addressed all my previous comments. My recommendation is that the manuscript is accepted for publication.

Dear editor,

We would like to thank the reviewers for their time and effort spent on the review process of our manuscript. We have revised our manuscript according to the referee's suggestions and highlighted the text modifications in blue. The line numbers refer to the tracked-change version of our revised manuscript.

Reviewer #1

Comments:

This manuscript reports on the structure and biochemical characterization of a novel member of the ferritin family, which has been named “mini-bacterioferritin” (Mper-mBfr). The authors isolated the mini-bacterioferritin from an enriched culture of the methanotrophic archaeon Candidatus Methanoperedens BLZ2. The crystal structure of the novel protein revealed a 12-mer protein assembled from identical subunits consisting of a 4-helix bundle that lacks the E-helix characteristic of bacterioferritins and the N' or C'-terminal extensions found in the subunits of Dps and Dps-like (DpsL) proteins. Unlike the 12-mer Dps, where the ferroxidase ligands are provided by two subunits in a dimer, the ferroxidase ligands in Mper-Bfr are contained within each subunit and are identical in composition to the ferroxidase ligands in bacterioferritin. Although these characteristics would suggest a 12-mer DpsL, the Mper-mBfr harbors a conserved set of Met residues (M50 in Mper-mBfr) which function as axial ligands to coproheme, which are attributes not found in DpsL. Taken together, the evidence supports a novel member of the ferritin family.

The manuscript articulates the findings clearly. A few issues should be addressed before the manuscript can be accepted for publication.

1. *The discussion indicates that Mper-mBfr probably serves as an iron storage protein, but the manuscript presents no evidence supporting this highly relevant idea. If Mper-mBfr indeed stores iron in its interior cavity, it is probable that the protein was isolated with an iron core. This can be readily determined in at least two ways, as has been shown for natively isolated Bfr from Pseudomonas aeruginosa and from Acinetobacter baumannii.*
 - a. *The UV-vis spectrum of Mper-mBfr encompassing the wavelength range 280-800 nm should be included in the manuscript. The presence of iron in the core of Mper-mBfr should give rise to an intense feature near 300 nm.*
 - b. *The presence of iron in the Mper-mBfr core can also be visualized in a native PAGE gel stained with Ferene.*
 - c. *If an iron core is not observed in the as isolated protein, attempts to reconstitute an iron core should be carried out in vitro, followed by analysis of the iron content to determine whether iron can be accumulated inside Mper-mBfr.*

(Our response) We thank the referee for the comments on our manuscript. We agree that there was no clear evidence presented for iron storage in the protein core. We have included a graph of the complete spectra in Figure 1B and the supplementary data (Supplementary Fig. S3). In addition, the raw data can be found in the Supplementary_Data_2.xlsx. In the as-isolated spectrum, we observed a peak around 310 nm that could be indicative of the presence of iron in the protein core. The peak seems to disappear upon oxidation and is unfortunately hidden by the dithionite signal upon reduction. We adjusted the text as follows:

Line 105: “A small peak around 310 nm might indicate the presence of Fe³⁺ (Fig. 1B) ²⁸⁻³¹.”

Line 383: “Based on the observed small peak around 310 nm in the UV-visible spectrum and the Fe shift in its active site upon oxidation, *Mper*-mBfr might function as an iron-storage protein, similar to *Dd*-Bfr¹³”

Supplementary Fig. S3. Complete spectra of *Mper*-mBfr in crystallo. *In crystallo* spectrum of the as-isolated crystal (pink), partially oxidised upon 10 minutes O₂ exposure (cyan), and oxidised upon 10 minutes O₂ exposure (green), then reduced by soaking the crystals for 4 min 40 sec in 100 mM sodium dithionite. Spectra are shown with applied baseline correction.

2. *Figure 6 and the accompanying text indicate that the di-iron ferroxidase center is in the ferrous oxidation state. Exposure to oxygen leads to loss in electron density from site Fe2, with new electron density developing at a new site, also labeled Fe2. The electron density attributed to iron in the structures should be corroborated by acquiring diffraction data near iron's absorption edge to calculate anomalous difference maps.*

We adjusted Figure 6 by including anomalous difference maps. The anomalous map corroborates the modelled occupancy (i.e., 40/60%) for alternative sites. Only very weak anomalous signals are detected in the core, which might be Fe at very low occupancy.

Line 229: “**Fig. 6. Redox cycling of *Mper-mBfr* crystals.** (A) *In crystallo* spectrum of the as-isolated crystal, partially oxidised upon 10 minutes O_2 exposure, and oxidised upon 10 minutes O_2 exposure then reduced by soaking the crystals for 4 min 40 sec in 100 mM sodium dithionite. The inset displays photos of the corresponding crystals. (B) Overlay of a dimer of each structure, with as-isolated in pink, oxidised in cyan and re-reduced in dark green. The inset shows an overlay of the active site of all three conditions. (C-E) Zoom-in of the active site for all three conditions. Bottom panel displays a 90° rotation along the x-axis of the top panel. The dark blue and orange mesh represents the $2F_o-F_c$ map contoured to 2.5σ for waters only, and the anomalous map contoured to 9σ , respectively. Ligands are represented as balls and sticks, with carbon, oxygen, nitrogen, and iron coloured in white, red, dark blue, and orange, respectively. Dashes display close interaction between the Fe and their surroundings. In panel D, interactions are only presented for Fe1 and the displaced Fe2.”

3. *Is the new Fe2 site observed upon oxidation closer to the protein interior, or to the protein exterior? Figure 6 should be modified to provide this context. The former would suggest that oxidation of the di-iron center leads to internalization of Fe(III).*

We now indicate the direction of the core in Figure 6. Upon oxidation, the Fe at site 2 is directed towards the solvent instead of the core. As the reviewer noted, this suggests that Fe(III) will not be transferred

into the central cavity for storage but instead delivered to the cytosol. Since this study is focused on the structure and phylogeny of the mBfr rather than its physiological function, we soften the discussion on the possible role of the protein.

Line 383: “Based on the observed small peak around 310 nm in the UV-visible spectrum and the Fe shift in its active site upon oxidation, *Mper*-mBfr might function as an iron-storage protein, similar to *Dd*-Bfr.¹³”

4. *Exposure of the crystals to oxygen for 10 minutes affects the Fe2 site but appears to leave the Fe1 site intact. The authors should explore longer exposure to oxygen to investigate the fate of Fe1. Will both iron ions in the di-iron center be lost from the ferroxidase center upon oxidation? This could indicate that the ferroxidase center in Mper-mBfr may act as a catalytic site and pore for Fe(III) internalization, as is the case in Bfr from P. aeruginosa.*

We agree that it would be interesting to investigate the mechanism of iron oxidation and internalisation in *Mper*-mBfr further. We attempted to crystallise the protein aerobically using all our available crystallisation kits, but obtained no hits. In contrast, anaerobic crystallisation resulted in a bloom of several crystalline forms. This could indicate that under aerobic conditions, there are conformational shifts in the active site that prohibit the formation of crystals in the tested conditions or might even lead to partial denaturation. It is, therefore, likely that longer exposure time of anoxic crystals to oxygen will lead to disintegration of the crystal or otherwise non-native conformational changes that might not reflect *in vivo* reaction mechanisms.

Reviewer #2

Comments:

In this work the authors describe a new class of ferritin like proteins. The overall findings and structural work presented is very exciting and of significant interest to the field. I'm excited to see this work published and only have a few minor points that should be addressed in the manuscript.

We thank the referee for the kind comments and appreciation of our results.

1. *In the introduction there is a description of the difference types of ferritin like biomolecules. However, there is no mention of the difference between heavy and light chain ferritins in the standard ferritins in the introduction. For example: " Ferritins oligomerise as 24-mer sphere-like structures (2,3,7–9). Each subunit contains a nuclear diiron ferroxidase centre that oxidises Fe²⁺ to Fe³⁺ using either oxygen or hydrogen peroxide (10). The iron is then mineralised and stored inside the hollow spherical cavity of the protein, which can hold up to 4,500 iron atoms (11)". Here it would be good to mention that the heavy chain contains the ferroxidase center while the light chain has a site for mineral nucleation. While I dont think it is important to this work to go too deep in to this, it should at least be mentioned for newcomers to the field of ferritin structure-function.*

The referee is right, ferritins can be composed of heavy and light chains in eukaryotes. To avoid confusion for the readers, we adjusted the sentence as follows:

Line 43: “While ferritins are composed of heavy and light chains in eukaryotes, they generally are homomultimers in prokaryotes, with each subunit containing a nuclear diiron ferroxidase centre that oxidises Fe²⁺ to Fe³⁺ using either oxygen or hydrogen peroxide^{9,10}.”

2. *In Figure 6D,E,F, it would be helpful to see this same view but with a lower contour to see the strength of the density around the oxygens. This could be shown in a supplemental figure.*

We adjusted Figure 6 by adding anomalous difference maps and placed a lower contour (2.5σ) $2F_o - F_c$ map around the oxygens of the water molecules coordinating the iron atoms. The previous Fig. 6 is now displayed as Supplementary Fig. S4 with an additional panel presenting the map at low contour.

Line 229: “**Fig. 6. Redox cycling of *Mper-mBfr* crystals.** (A) *In crystallo* spectrum of the as-isolated crystal, partially oxidised upon 10 minutes O_2 exposure, and oxidised upon 10 minutes O_2 exposure then reduced by soaking the crystals for 4 min 40 sec in 100 mM sodium dithionite. The inset displays photos of the corresponding crystals. (B) Overlay of a dimer of each structure, with as-isolated in pink, oxidised in cyan and re-reduced in dark green. The inset shows an overlay of the active site of all three conditions. (C-E) Zoom-in of the active site for all three conditions. Bottom panel displays a 90° rotation along the x-axis of the top panel. The dark blue and orange mesh represents the $2F_o - F_c$ map contoured to 2.5σ for waters only, and the anomalous map contoured to 9σ , respectively. Ligands are represented as balls and sticks, with carbon, oxygen, nitrogen, and iron coloured in white, red, dark blue, and orange, respectively. Dashes display close interaction between the Fe and their surroundings. In panel D, interactions are only presented for Fe1 and the displaced Fe2”

Supplementary Fig S4. Redox cycling of *Mper*-*mBfr* crystals. Zoom-in of the active site of the as-isolated (A), partially oxidised upon 10 minutes O_2 exposure (B), and partially oxidised upon 10 minutes O_2 exposure then reduced by soaking the crystals for 4 min 40 sec in 100 mM sodium dithionite (C) conditions. The top panel displays the distances between ligands in Å using dotted lines. The $2F_o-F_c$ map is contoured around the iron atoms to 7σ as a dark blue mesh. The middle panel displays the same view, but with the $2F_o-F_c$ map as a dark blue mesh contoured to 2.5σ around the iron atoms, the coordinating ligands, and the water molecules. The bottom panel displays a 90° rotation along the x-axis of the middle panel. Ligands are represented as balls and sticks, with carbon, oxygen, nitrogen, and iron coloured in white, red, dark blue, and orange, respectively.

- Regarding the coproheme discussed here: "The atomic resolution electron density map revealed a dual conformation of the coproheme ligand, with each orientation present at approximately 50% occupancy (Fig. 4A)" Was the occupancy refined during refinement of the model? When you say approximately 50%, was the model refined to 50%? I think it would be helpful to give the value it actually refined to instead of just saying 50%. If what is meant by the "approximately 50%" is that I wasn't completely clear, maybe mention that with a bit more detail in the text. With this quality of structure I would think you would be able to provide a more clear number, but maybe that isn't the case. either way, some more detail here would be helpful.

We agree that this statement was unclear. The structures were refined with exactly 50% occupancy, which led to the best b-factor values highlighted in Fig. 4C (i.e., similar in both model coproheme and fitting the surroundings). We clarified this in the text:

Line 183: The atomic resolution electron density map revealed a dual conformation of the coproheme ligand, with each orientation modelled at 50% occupancy, fitting at best the individual atomic b-factor profile (Fig. 4).

Dear editor,

We would like to thank both reviewers for their time and consideration of our work. We followed the first reviewer's recommendation and investigated whether the mini bacterioferritin could internalize Fe. The experiment showed that while the as-isolated protein does not contain Fe in its hollow core, it can incorporate it in the form of Fe(III) upon addition of Fe(II)Cl₂ and H₂O₂. This added experiment in the manuscript confirms that the enzyme retains ferritin activity *in vitro*.

Reviewer #1 (Remarks to the Author):

Comments

The authors have addressed most of the concerns raised in the original review. There is, however, the remaining issue of whether the novel ferritin-like protein (Mper-mBfr) can store iron in its interior cavity. This concern had been brought up in the previous review but was not properly addressed in the revised manuscript. This important issue, which is aimed at determining whether the novel Mper-mBfr protein exhibits the classical iron storage function must be properly dealt with before the manuscript can be accepted for publication. More detailed explanation and some suggestions are outlined below

(Our response) We agreed with the reviewer that this information is important for verifying whether the mini-bacterioferritin can store iron and investigated this possibility experimentally (see below).

The UV-vis spectrum in Figure 1 of the revised manuscript shows a small peak at 310 nm, which prompted the authors to speculate “might indicate the presence of Fe(III) (Fig. 1B)”. It is highly improbable that this peak originates from Fe(III)-O- moieties in the interior cavity; the presence of Fe(III)-O- gives rise to a broad, intense band which does not peak at 310 nm. Furthermore, the small 310 nm peak disappears in the presence of oxygen, providing further support for the idea that the small peak does not originate from Fe(III)-O-. Rather, the small 310 nm peak is probably a feature of the Fe-porphyrin system. Consequently, the spectra in Fig. 1 indicate that the protein, as isolated, is devoid of an iron core, and the authors should indicate this in the results and discussion.

We thank this expert for helping us to interpret the UV/Visible spectra and use this explanation to modify the text accordingly. Moreover, our biochemical experiment confirms the absence of Fe(III) in the as-isolated state of the mini-bacterioferritin, corroborating the reviewer's view.

Lines 106-111: “A small peak at 310 nm is detected in the as-isolated form (Fig. 1B), however, it is highly improbable that this peak originates from Fe(III)-O- moieties in the interior cavity since the presence of Fe(III)-O- is expected to give rise to a broad intense band near 300 nm²⁸⁻³¹. Furthermore, this peak disappears in the presence of oxygen, providing further support that it does not originate from Fe(III)-O-, and would rather be a feature of the Fe-porphyrin system.”

In the revised manuscript the authors made no attempt to reconstitute an iron core in vitro, despite the concern having been raised in the previous review (l.c.): “If an iron core is not observed in the as isolated protein, attempts to reconstitute an iron core should be carried out in vitro, followed by analysis of the iron content to determine whether iron can be accumulated inside Mper-mBfr”.

Reconstitution of an iron core in vitro is an important and commonly performed biochemical assay carried out with ferritin and ferritin-like proteins to determine whether the ferroxidase centers are functional and if the protein accumulates iron in its interior. Consequently, attempts to reconstitute an iron core in the novel Mper-mBfr should be performed and the results reported in the manuscript. This assay can be carried out anaerobically in a titration where the addition of each Fe(II) aliquot is followed by the addition of an equivalent of H₂O₂ (e.g. ref. 21). Reconstitution of an iron core would indicate that one of the functions of the novel Mper-mBfr is in the detoxication of Fe(II) and storage of Fe(III)

(the classical iron storage function) which is one role attributed to most Dps and Dps-like proteins. On the other hand, if an iron core cannot be reconstituted *in vitro*, together with the fact that the isolated protein is devoid of an iron core, it would suggest an entirely different function, such as a possible role in the mitigation from H₂O₂-mediated stress.

We performed the recommended experiment by first establishing the assay using commercial horse ferritin and then applying it to a fresh, enriched mini-bacterioferritin batch. Supplementary Figure 5 describes the presence of Fe(III) in the native horse ferritin, and when the sample is treated with Fe(II)Cl₂/H₂O₂ treatment. Moreover, the Fe signal disappears when the ferritin is directly treated with dithionite. Regarding the mini-bacterioferritin, Fe(III) is detected only after Fe(II)Cl₂/H₂O₂ treatment, indicating its absence in the as-isolated state. This absence might result from the mild reducing conditions used during protein extraction and purification (an anaerobic environment with 2 mM dithiothreitol and natural reducing molecules from the microbial extract).

We integrated these results into the manuscript as follows:

Lines 259-269: “The observation of the displacement of Fe²⁺ upon redox cycling prompted us to investigate if *Mper*-mBfr harbours Fe-storing activity. For this, we performed Ferene S and Prussian Blue staining on hrCN-PAGE to detect Fe- and Fe(III)-containing ferritin, respectively (Supplementary Fig. S5). In the positive control using horse ferritin, Fe(III) is detected in the native state, and upon incubation with Fe(II)Cl₂/H₂O₂. However, the signal disappears when the ferritin is directly reduced by dithionite. In the case of *Mper*-mBfr, the Fe(III) is only detected when the protein is incubated with Fe(II)Cl₂/H₂O₂. Thus, the experiment confirms that *Mper*-mBfr exhibits the classical ferritin activity of Fe internalisation *in vitro*. The absence of Fe(III) in the as-isolated state might be due to the mild-reducing conditions during protein extraction and purification (e.g., anaerobic conditions and the presence of dithiothreitol and natural reducing agents from the microbial community).”

Horse ferritin

Mper-mBfr

Supplementary Fig. 5. Supplementary Fig. S5. Ferritin activity and Fe incorporation. Commercial horse ferritin and enriched *Mper*-mBfr were incubated for 45 minutes at 20 °C anaerobically in the presence of dithionite or Fe(II)Cl₂/H₂O₂ to assay their capacity for internalising Fe. 4.5 µg of protein were loaded on hrCN PAGE. After migration, the gel was cut into three parts. Each part was stained with a specific solution: Ready Blue to detect all proteins, Ferene S to detect Fe, and Prussian Blue to detect Fe(III). *Mper*-mBfr enrichment protocol via ammonium sulfate precipitation resulted in slight contamination, as evidenced by two additional bands between 20 and 66 kDa.

Lines 404-407: “The physiological function of *Mper*-mBfr remains speculative at this stage. Based on the Fe internalisation experiment (Supplementary Fig. 5), we concluded that the enzyme harbours ferritin activity *in vitro* and might function as an iron-storage protein *in vivo*, similar to *Dd*-Bfr.¹³”

Reviewer #2 (Remarks to the Author):

The authors have addressed my small concerns in the initial submission. I would support publications of this work.

We are delighted that Reviewer 2 supports the publication and appreciates our work.